# ORACLES & FOLLOWERS: STACKELBERG EQUILIBRIA IN DEEP MULTI-AGENT REINFORCEMENT LEARNING

## ABSTRACT

Stackelberg equilibria arise naturally in a range of popular learning problems, such as in security games or indirect mechanism design, and have received increasing attention in the reinforcement learning literature. We present a general framework for implementing Stackelberg equilibria search as a multi-agent RL problem, allowing a wide range of algorithmic design choices. We discuss how previous approaches can be seen as specific instantiations of this framework. As a key insight, we note that the design space allows for approaches not previously seen in the literature, for instance by leveraging multitask and meta-RL techniques for follower convergence. We propose one such approach using contextual policies and evaluate it experimentally on standard and novel benchmark domains, showing greatly improved sample efficiency compared to previous approaches. Finally, we explore the effect of adopting designs outside the borders of our framework.

## 1 INTRODUCTION

Stackelberg equilibria are an important concept in economics, and in recent years have received increasing attention in computer science and specifically in the multiagent learning community. In these equilibria, we have an asymmetric setting: A leader who commits to a strategy, and one or more followers who respond. The leader aims to maximize their reward, knowing that followers in turn will best-respond to the leader's choice of strategy. These equilibria appear in a wide range of settings. In security games, a defender wishes to choose an optimal strategy considering attackers will adapt to it (An et al., 2017; Sinha et al., 2018). In mechanism design, we aim to design a mechanism that allocates resources in an efficient manner, knowing that participants may strategize (Nisan & Ronen, 1999; Swamy, 2007; Brero et al., 2021a). More broadly, many multi-agent system design problems can be viewed as Stackelberg equilibrium problems: we as the designer take on the role of the Stackelberg leader, wishing to design a system that is robust to agent behavior.

We are particularly interested in Stackelberg equilibria in sequential decision making settings, i.e. stochastic Markov games, and using multi-agent reinforcement learning techniques to learn these equilibria. In this paper we:

1. Introduce a new theoretical framework for framing Stackelberg equilibria as a multi-agent reinforcement learning problem,

2. Discuss how a range of existing approaches fit into this paradigm, as well as where there remain large unexplored areas in the design space,

3. Reveal a novel approach to accelerating follower best-response convergence, borrowing ideas from multitask and meta-RL, including an experimental evaluation, and

4. Elaborate on several important conditions for Stackelberg convergence, and demonstrate how things can fail when these conditions are not met.

Our main theorem (Theorem 1) allows a black-box reduction from learning Stackelberg equilibria into separate leader and follower learning problems. This framing encompasses and generalizes several prior approaches from the literature, in particular Brero et al. (2022), and gives a large design space beyond what has been explored previously. Our second main technical contribution is applying contextual policies, a common tool in multitask and meta-RL (Wang et al., 2016), to the follower learning problem. In doing so, followers can generalize, and quickly adapt to leader

policies. We validate this approach in experiments and show greatly reduced sample complexity compared to previous inner loop-outer loop approaches. We also show how violating the conditions of our theorem can lead to complete failure of the learning process, consistently across underlying algorithms.

In the remainder of the paper, we will introduce Stackelberg equilibria and Markov games in Section 2. In Section 3, we motivate and define our framework for learning Stackelberg equilibria using multi-agent RL and discuss its scope and limitations. We define our novel contextual policy oracle in 4 and empirically evaluate it in Section 5 on existing and novel benchmark domains.

## 1.1 PRIOR WORK

**Learning Stackelberg Equilibria.** Most prior work on Stackelberg equilibria focus on single-shot settings such as normal-form games, a significantly simpler setting than Markov games, often in security games. A broad line of work focuses on *computing* Stackelberg equilibria, such as Paruchuri et al. (2008); Xu et al. (2014); Blum et al. (2014); Li et al. (2022). Among the first works on *learning* Stackelberg equilibria was Letchford et al. (2009), who focus on Bayesian games. Peng et al. (2019) give results for matrix games with sample access only. Wang et al. (2022) show an approach differentiating through the so-called KKT conditions, again for normal-form games. Bai et al. (2021) give lower and upper bounds on learning Stackelberg equilibria in general-sum games, including "bandit RL" games with one step for the leader, and sequential decision-making for the followers. Few works in this area consider Markov games: Zhong et al. (2021) show algorithms that find Stackelberg equilibria in Markov games, but assume myopic followers, a significant limitation compared to the general case. Brero et al. (2021a;b; 2022) use an inner-outer loop approach, which they call the *Stackelberg POMDP*, primarily aimed at indirect mechanism design.

**Mechanism Design.** One of the first works specifically discussing Stackelberg equilibria in a learning context is Swamy (2007), who design interventions in traffic patterns. More recently, several strands of work have focused on using multi-agent RL techniques to learn optimal mechanism design, often framing this is a bi-level or inner-outer-loop optimization problem. Zheng et al. (2022) use a bi-level RL approach to design optimal tax policies in a simulated world but without provably Stackelberg properties. Yang et al. (2022) use a meta-gradient approach in a specific incentive design setting. Shu & Tian (2018) and Shi et al. (2019) learn leader policies in a similar "Stackelberg Markov games" setting. Both use a form of modeling other agents coupled with rule-based followers. Balaguer et al. (2022) use an inner-loop outer-loop gradient descent approach for mechanism design on iterated matrix games (which we also use as an experimental testbed). They mainly focus on the case where both the environment transition as well as the follower learning behavior is differentiable, and otherwise fall back to evolutionary strategies for the leader. Interestingly, none of these recent works explicitly mention Stackelberg equilibria. As a direct corollary of our own work, we show that Balaguer et al. (2022) and Zheng et al. (2022) may not give Stackelberg equilibria, but Balaguer et al. (2022) could do so with minor modifications.

## 2 PRELIMINARIES

**Markov games.** We consider partially observable stochastic Markov games, essentially a multi-agent generalization of a partially observable Markov Decision Process (POMDP).

**Definition 1** (Markov Game). *A Markov Game $\mathcal{M}$ with $n$ agents is a tuple $(S, A, T, r, \Omega, O, \gamma)$, consisting of a state space $S$, an action space $A = (A_1, ..., A_n)$, a (stochastic) transition function $T : S \times A \to S$, a (stochastic) reward function $r : S \times A \to \mathbb{R}^n$, an observation space $\Omega = (\Omega_1, ..., \Omega_n)$, a (stochastic) observation function $O : S \times A \to \Omega$, and a discount factor $\gamma$.*

At each step $t$ of the game, every agent $i$ chooses an action $a_{i,t}$ from their action space $A_i$, the game state evolves according to the joint action $(a_{1,t}, \ldots, a_{n,t})$ and the transition function $T$, and agents receive observations and reward according to $O$ and $R$. An agent's behavior in the game is characterized by its policy $\pi_i : o_i \mapsto a_i$, which maps observations to actions.[1] Each agent in a Markov

---

[1]To keep notation concise we discuss here the memory-less case, but all our results generalize to a stateful leader policy in a straightforward manner, as we discuss in Appendix B.

Game individually seeks to maximize its own (discounted) total return $\sum_t \gamma^t r_i(s_t, a_{i,t}, a_{-i,t})$. This gives rise to the usual definitions of Nash equilibria (NE), correlated equilibria (CE), and coarse correlated equilibria (CCE), which we do not repeat in full here, as well as their Bayesian counterparts. Note that strategies in Markov games and in each of these equilibrium definitions are *policies*, not actions: A pair of policies $\pi_1, \pi_2$ in a two-player Markov game is a Nash equilibrium if neither agent can increase their expected total reward by unilaterally deviating to a different policy $\pi_i'$.

**Stackelberg Equilibria.** Unlike all the above equilibrium concepts, a Stackelberg equilibrium is not symmetric: There is a special player, the leader, who commits to their strategy first; the other player (the follower) then chooses their best strategy given the leader's choice of strategy. This makes the leader potentially more powerful.

**Example 1.** *In a game often called the "battle of the sexes," you and I wish to have dinner together, but you prefer restaurant A (deriving happiness 2, but I only get happiness 1), and I prefer restaurant B (I get happiness 2, you get happiness 1)—but we would both rather eat together at our less-preferred venue, than to eat separately (we both get happiness 0). Table 1 shows the payoff matrix of this game. There are two pure Nash equilibria in this game: We both go to restaurant A, or we both go to restaurant B. But there is only a single Stackelberg equilibrium (per leader): If you commit to going to restaurant A, then my only best response is to also go to restaurant A. In doing so I receive happiness 1, whereas my only alternative would be to eat alone at restaurant B for happiness 0. Notice that this hinges on the leader strictly committing to their choice of restaurant.*

Table 1: "Battle of the Sexes" game.

| 2,1 | 0,0 |
|-----|-----|
| 0,0 | 1,2 |

This Stackelberg concept again also extends to Markov games: Here a leader agent $L$ decides on their policy (i.e. strategy), and the remaining (follower) agents — knowing the leader's choice of policy — best-respond to this. The leader seeks to maximize their own reward, considering that followers will always best-respond to their choice of policy. For instance, in an iterated prisoner's dilemma (Robinson & Goforth, 2005), a leader might commit to a tit-for-tat strategy, in turn leading the follower to cooperate.

We are interested in multi-follower settings, where Stackelberg equilibria have also been defined Nakamura (2015); Zhang et al. (2016); Liu (1998); Solis et al. (2016); Sinha et al. (2014); Wang et al. (2022); Brero et al. (2021a). With multiple followers, the "best response" of the followers can be an ambiguous term. Any choice of leader strategy $\pi_L$ induces a Markov game between $\mathcal{F}_{\pi_L}$ between the followers, which could feature multiple equilibria and equilibria of different types, such as Nash, correlated and coarse correlated equilibria, each giving rise to a corresponding Stackelberg-Nash, Stackelberg-CE and Stackelberg-CCE concept. We handle this in our definition through an oracle (see also e.g. Wang et al. (2022): For any choice of leader strategy, we assume we are given a follower equilibrium (or a probability distribution over equilibria), $\mathcal{E}(\mathcal{F}_{\pi_L})$, by an oracle $\mathcal{E}$.

**Definition 2** (Stackelberg equilibrium). *Given a Markov Game $\mathcal{M}$ and a follower best-response oracle $\mathcal{E}$, a leader strategy $\pi_L$ together with a tuple of follower strategies $\pi_F$ is a Stackelberg equilibrium, if $\pi_L$ maximizes the leaders expected reward under the condition that follower strategies are drawn from $\mathcal{E}(\mathcal{F}_{\pi_L})$:*

$$\pi_L \in \arg\max_{\pi_L} \mathbb{E}_{\pi_F \sim \mathcal{E}(\mathcal{F}_{\pi_L})} \left[ \sum_t \mathbb{E}[r_L(s_t, a_{L,t}, a_{F,t})] \right], \tag{1}$$

*where the second expectation is drawing actions and state transitions from their respective policies $\pi_L, \pi_F$ and transition function $T$, and the reward function is $r$, all as in Definition 1. If the follower oracle $\mathcal{E}$ gives a Nash equilibrium in the induced game $F_{\pi_L}$, we call this a Stackelberg-Nash equilibrium, and similar for CE and CCE.*

This use of an oracle is convenient not only for the multiple-follower definition, but also a crucially useful abstraction for any number of followers. What we have in mind for practical purposes is an algorithm that computes or learns the follower best-response equilibrium, $\mathcal{E}(\mathcal{F}_{\pi_L})$, given the leader strategy $\pi_L$, often using RL. In the remainder of the paper, when we say "oracle" this is what we mean.

## 3 A GENERAL FRAMEWORK FOR STACKELBERG EQUILIBRIA IN MULTI-AGENT RL

Several approaches have been proposed to learning Stackelberg equilibria in Markov games, or to use multi-agent RL for mechanism design in such settings. A main aim of our work is to elucidate commonalities between these approaches, and to delineate what is required to guarantee Stackelberg equilibria. For instance, most of the existing approaches use (reinforcement or no-regret) learning to implement the follower best-response, effectively arriving at an *"inner-loop-outer-loop system"*: The leader performs one update to their policy, then the followers perform many updates to theirs until they converge to a best response, then this repeats. Is this the only possible approach? Can you mix-and-match leader and follower approaches at will? One approach for leader learning is reinforcement learning, where the gradient of the leader policy is estimated from sampled trajectories (Brero et al., 2021a)—this is in contrast to global approaches such as direct differentiation of the leader policy in a world where everything is differentiable or evolutionary strategies (Balaguer et al., 2022), which modify the leader policy as a whole based on total episode reward, without looking at what happens at each step. Some RL approaches (Brero et al., 2021a; 2022) incorporate the followers' learning dynamics into the leader's view of the environment. Is it necessary that the leader can see this adaption process? Or could we also have follower best-respond to the leader immediately on the first step it takes? Some approaches for mechanism design do not explicitly mention Stackelberg equilibria (Zheng et al., 2022; Balaguer et al., 2022), but seem very similar to approaches that do; do those approaches converge to a Stackelberg equilibrium? In this section we develop a common framework that answers these questions, and provide a common language to categorize the various strands of research in this area.

A key idea is that we can separate the problem into a leader's learning problem, and an implementation of the follower oracle. Our framework informally, provides two insights. One, for *any* follower oracle implementation and *any* construction of the leader problem, a solution is a Stackelberg equilibrium as long as the reward given to the leader is that from the original Markov game with best-responding followers. A key insight following from this is that follower oracles need not be implemented using learning algorithms, leading to a new approach which we will outline in Section 4. This also encompasses a wide range of leader learning approaches, including direct gradient descent, optimization, and evolutionary strategies. At the same time it gives very simple conditions to ensure Stackelberg solutions, and delineates approaches that do and do not give these. Two, for any query-based oracle implementation, we can construct the leader problem as a POMDP, amenable to solving through RL methods. Since most follower oracle implementations are query-based (or can be approximated through queries), this again encompasses a wide range of possible approaches. Together, these simple conditions open a wide design space, which includes existing approaches but also provides a much wider field of possibilities, one of which we demonstrate in Section 4.

We now make this formal. The general case of the following theorem is intentionally kept close to the definition of a Stackelberg equilibrium, and we will discuss implications of this after the theorem statement.

**Theorem 1.** *Given a Markov Game $\mathcal{M}$ and a follower equilibrium oracle $\mathcal{E}$, then:*

***[General Case]** If*

1. *the leader locally aims to solve a single-agent optimization problem $\mathcal{L}(\pi_L)$, and*

2. *for each choice of leader policy $\pi_L$, $\mathcal{L}$ computes the follower best-response $\mathcal{E}(\pi_L)$, and*

3. *$\mathcal{L}(\pi_L)$ evaluates the leader policy $\pi_L$ against the follower best-response $\mathcal{E}(\pi_L)$ in $\mathcal{M}$, i.e. the value of $\mathcal{L}(\pi_L)$ is $r_L(\pi_L, \mathcal{E}(\pi_L))$ in $\mathcal{M}$,*

*then an optimal solution $\pi_L^*$ to $\mathcal{L}$ together with the follower best-response $\mathcal{E}(\pi_L^*)$ form a Stackelberg equilibrium in $\mathcal{M}$.*

***[Query-Oracle Special Case]** Furthermore, if additionally*

4. *the follower oracle implementation $\mathcal{E}$ only requires query access to $\pi_L$, i.e. values $\pi_L(o)$ for one or more observations $o \in \Omega_L$ from $\mathcal{M}$,*

*then $\mathcal{L}$ can be constructed as a POMDP.*

*Proof.* See Appendix A. We briefly note for the following discussion that the construction in the query-oracle case constructs episodes in $\mathcal{L}$ by using an *initial segment* in which the oracle queries $\pi_L$, and a *final segment* in which an episode of $\mathcal{M}$ is played by $\pi_L$ and the $\mathcal{E}(\pi_L)$ computed in the initial segment. This is formalized in the proof in the Appendix. □

Theorem 1 unifies several approaches from the literature. The main theorem of Brero et al. (2021a) can be seen as a special case of the query-oracle case. However, even the query-oracle special case of our theorem significantly generalizes that result to allow for *any* query-based follower oracle. Furthermore, the general case of the theorem extends this to approaches that do not require the leader problem to be a POMDP. While the general case follows very easily from Definition 2, it still provides a very powerful insight. For instance, the "good shepherd" approach (Balaguer et al., 2022) performs gradient descent directly on the leader policy, either by differentiating through the environment and followers, or by estimating the gradient using evolutionary strategies. Neither requires the problem to be a POMDP. At the same time, the general case of Theorem 1 applies, and it tells us that the "good shepherd" approach in its current form may not give Stackelberg equilibria: it accumulates leader reward even during follower learning, i.e., when followers are not (yet) playing $\mathcal{E}(\pi_L)$. This may be intentional: In the "good shepherd" approach the leader optimizes its expected return over a longer horizon during which followers adapt from scratch. This is a subtly different optimization target than a Stackelberg equilibrium, which optimizes steady-state return (see also Appendix C.3). Further, the theorem does not make any assumptions about the type of access to the environment and follower oracle, and can thereby encompass approaches that only require sample access as well as those that have access to a description, and approaches where followers or the environment are differentiable.

### 3.1 SOLVING THE LEADER PROBLEM

Theorem 1 does not assume a specific approach to solve the leader problem $\mathcal{L}$; rather, it gives us a broad array of tools to solve Stackelberg problems in practice. One particular set of tools lie in the query-oracle special case: by constructing $\mathcal{L}$ as a POMDP, the entire suite of existing (deep) RL algorithms become available. We later show experimental results using PG, PPO and DQN, but this is by no means exhaustive.

A reasonable question to ask is whether the POMDP property is actually required for RL algorithms to solve these problems. For instance, if we implemented the follower oracle $\mathcal{E}$ in a way that is not part of the leader problem $\mathcal{L}$, essentially skipping the "initial segment" used in the construction of $\mathcal{L}$ in the special case of Theorem 1, would RL still be able to solve the leader problem? Theorem 2 in Appendix D shows that the POMDP property of the query-oracle construction is meaningful in practice: There are Markov games where an RL algorithm provably cannot learn the optimal leader policy if the follower best-response oracle is not implemented in this manner. This is further demonstrated experimentally in Appendix C.1.

Appendix B further details how to extend Theorem 1 to leader policies with memory, and details a crucial condition of *leader invariance* which we would like to draw particular attention to. This condition requires that the leader policy act the same during oracle queries as it does during real play. In the memory-less case this follows immediately in theory, but is an important implementation detail in practice. In Appendix C.4 we show an example where a seemingly innocuous step counter being made part of the leader's observation leads to learning failure.

### 3.2 IMPLEMENTING THE FOLLOWER ORACLE

A key insight is that Theorem 1 works for *any* implementation of a follower oracle. Some of the prior literature focuses entirely on the leader problem and assumes a literal oracle for follower best-response, sometimes also requiring this be differentiable. Practical follower oracle implementations in the literature can broadly be grouped into two categories.

**Optimization-based approaches**, which have access to a description of the environment (e.g. a payoff matrix), have been used in simple one-shot settings such as (non-iterated) matrix games, where a description of the leader policy and environment can be used to compute an optimal follower response. This is often coupled with optimization for the leader policy, for instance in Paruchuri et al. (2008); Xu et al. (2014).

**Learning-based approaches**, which require only sample access to the environment, and potentially the leader policy. This approach is taken in some of the single-shot Stackelberg literature and most of the RL-based Stackelberg literature: every time the leader policy is updated to a new policy, $\pi_{L,\text{new}}$, followers use a standard learning algorithm, for instance RL or regret minimization in the Markov game case, to learn in the induced game $\mathcal{F}_{\pi_{L,\text{new}}}$. Any convergence guarantee for the follower algorithm (e.g., CCE in no-regret, or Nash in V-Learning in some conditions) translates to the equivalent Stackelberg-CCE or Stackelberg-Nash equilibrium. Follower weights can be re-initialized randomly after each leader update, which makes the environment stationary, or trained without this (possibly requiring less training, in the case the leader policy only changed a little, but see Appendix C.4). It is worth noting that our framework using an RL oracle "looks like" leader and follower are simultaneously learning to act in a typical multi-agent RL system, just at different timescales. However we find it useful to think of this setup in terms of leader learning as separate from follower oracle, and there are important differences from standard multi-agent RL: The leader should receive reward only when followers are playing their best-response equilibrium, and the entire follower learning process should be one episode for the leader.

## 4    META-RL FOR STACKELBERG RL

Going beyond this, Theorem 1 suggests a wide design space for implementing the follower oracle. As a key contribution, we explore using multi-task and meta-RL as a means of implementing the follower oracle. This is both to illustrate the power of Theorem 1 as a way to think about Stackelberg learning, as well as due to the advantages of the approach over existing ones.

We can recognize that the follower games, $\mathcal{F}_{\pi_L}$, are in fact a family of related problems. For this reason, the follower oracle problem can be seen as a multitask or meta-RL problem, and solved using techniques from those fields. We make use of *contextual policies* (Wang et al., 2016), where a context $\omega$ describes the task an agent is supposed to solve. In our case, the context provides the specific MDP among a family of MDPs a follower finds itself in, and $\omega$ is a description of the leader policy. This context, $\omega$, is concatenated to the follower agent's observation $o_{i,t}$, and agent $i$ observes $(o_{i,t}, \omega)$ at timestep $t$.

We focus on settings where the leader policy's effect on the follower can be fully understood with a small number of queries, and we directly use the leader's response to a fixed set of queries as the context $\omega$. For instance, in an iterated prisoner's dilemma, we ask the leader three questions: "How do you act on the initial step of the game?", "How do you act if the opponent cooperated in the previous step?" and "How do you act if the opponent defected in the previous step?" Clearly, if these are the only three possible states, this is sufficient to characterize the leader policy. We further use a two-stage training approach. In Phase 1, we train a follower meta-policy against a different, randomized leader policy in each episode. By the end of this phase, the meta-policy is able to best-respond to all possible leader policies. In Phase 2, we train a leader policy against this follower, where the leader is queried at the beginning of each episode. In our current experiments we explicitly define the context $\omega$ in the above way. For settings where this is not possible, the multitask and meta-RL literature provides a range of approaches that infer context, often using recurrent networks (Wang et al., 2016; Mishra et al., 2017; Duan et al., 2016; Rakelly et al., 2019; Zintgraf et al., 2019; Humplik et al., 2019).

**Relation to existing approaches in the literature**    Previous approaches for Markov games have largely focused on no-regret and policy gradient learning to implement follower oracles, coupled with either RL or direct gradient descent methods for the leader. Brero et al. (2021a; 2022) use no-regret dynamics or Q-learning to implement the follower oracle inside the leader's episode roll-out, and standard RL techniques to solve the resulting leader POMDP. Balaguer et al. (2022) use gradient methods to implement the follower oracle. In the case where both followers and world dynamics are differentiable they directly differentiate the leader policy (as opposed to estimating its gradient using sampled trajectories). For the non-differentiable case they use evolutionary strategies (Salimans et al., 2017). Interestingly, they seem to accumulate leader reward throughout the entire learning phase of the follower. This puts their approach outside the scope of our Theorem 1, and may give the leader the wrong optimization target, as we detail in Appendix C.3. Our understanding is that if the leader were to optimize for its final reward at the end of follower learning instead, their approach would fall within Theorem 1 and yield Stackelberg equilibria. Zheng et al. (2022)

Table 2: Situating different approaches within the framework of Theorem 1. Approaches marked * do not fully satisfy the conditions of the theorem. Approaches marked in bold indicate approaches that apply to general Markov games, i.e., that are sequential for both leader and followers, and otherwise unrestricted.

| | Leader Learning Approach | | |
|---|---|---|---|
| *Oracle Implementation* | Optimization, Search | Direct Gradient Descent, Evolutionary Strategies | RL |
| N/A - Oracle Assumed Given | Letchford et al. (2009) Peng et al. (2019) | Wang et al. (2022) | Zhong et al. (2021) |
| No-Regret | | | Brero et al. (2021a) |
| RL | Bai et al. (2021) | **Balaguer et al. (2022)\*** Yang et al. (2022)\* | **Zheng et al. (2022)\*** **Brero et al. (2022)** |
| Multitask / Meta-RL | | | **new** |

similarly use two-level RL design, using policy gradient to learn each of the follower oracle and the leader policy. While they do not specifically mention Stackelberg equilibria, this would be the target equilibrium condition in their taxation-policy design setting. Interestingly, they use a curriculum learning approach that can be seen as a rudimentary form of the contextual policy meta-learning approach that we develop in this paper. As with Balaguer et al. (2022), Zheng et al. (2022) also seem to accumulate leader reward, putting them outside the assumptions of Theorem 1. It is also not clear if their implementation of leader memory could violate the leader invariance requirement. However, it may be possible to adapt their approach to fall within the theorem by adjusting leader reward and resetting leader memory between follower queries / learning iterations.

## 5 EXPERIMENTS

We evaluate our Meta-RL approach on both a benchmark iterated matrix game domain, comparing to existing approaches, as well as on a novel Atari 2600-based domain that is significantly more challenging. In the first, our main positive finding is that our approach can match or exceed prior approaches at greatly improved sample efficiency. In the latter, we show for the first time a positive result using a principled Stackelberg approach on a state-of-the-art general RL benchmark domain. We will detail each of the two domains and the results obtained therein using the Meta-RL algorithm. Appendices F and G give further details on the algorithms used and full hyperparameters.

**Environments: Iterated Matrix Games** We evaluate our contextual policy approach and general framework on an ensemble of iterated symmetric matrix games, such as the iterated prisoner's dilemma (Robinson & Goforth, 2005). We choose these games as an evaluation domain as they present a significant step up in complexity from previous approaches that give explicit Stackelberg guarantees, in that both leader and followers face a sequential decision-making problem. In these, we play a matrix game for $n = 10$ steps per episode, and give agents a one-step memory. This makes these environments Markov games, with five states: one for the initial steps of each episode, and four for later steps depending on the two agents' previous actions. At each step, each agent has a choice of two actions (e.g. "cooperate" or "defect"), leading to the next state, e.g. "both cooperated".

Figure 1 shows the performance of our Meta-RL approach using PPO for the leader. We compare against the approaches of Balaguer et al. (2022) and Brero et al. (2022). For our PPO+Meta-RL approach, we plot the combined environment steps used by the meta-follower training plus the leader training on the x-axis. For Balaguer et al. (2022), we estimate performance from Figure 2 therein. Note that this is the eventual performance at the end of training (Balaguer et al. (2022) do not publish learning curves).

We see that final performance largely matches that of the "good shepherd" ES-MD approach, and by extension also matches or outperforms all their baselines (c.f. Figures 1 and 2 therein). Importantly, notice that our approach converges in around 50k environment steps, whereas Balaguer et al. (2022)

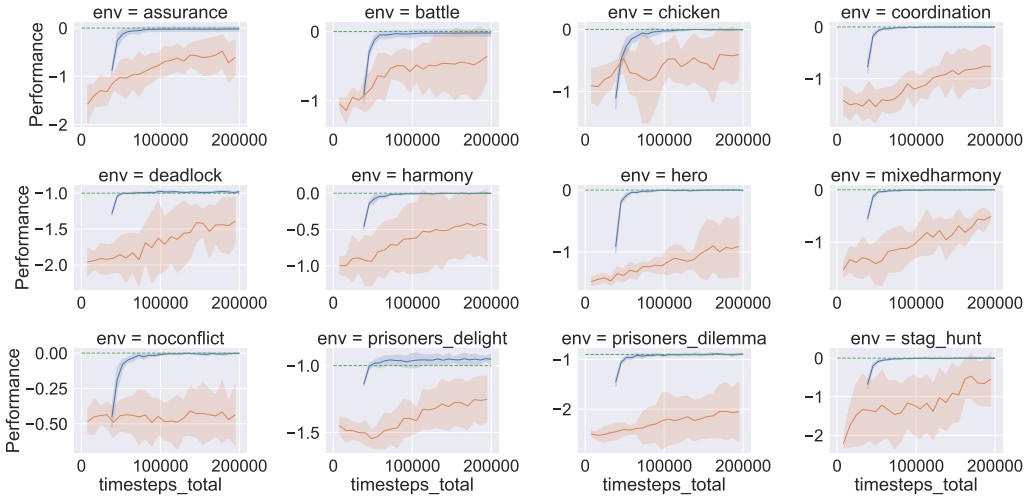

Figure 1: Blue: Performance of our novel PPO+Meta-RL approach on 12 canonical symmetric iterated matrix games. Orange: PPO+Q-learn Brero et al. (2022). Dashed green: Good Shepherd ES-MD Balaguer et al. (2022) (final performance at 1.28B timesteps, estimated from Fig. 2 ibid.)

report performance at 1.28 billion environment steps in the ES-MD case. We give further details on comparing performance to Balaguer et al. (2022) in Appendix H.

We also see that our approach outperforms the PPO+Q-learn approach of Brero et al. (2022). In Appendix H we show the PPO+Q-learn approach training for significantly longer, and see that where it does converge it does so around 500k environment steps at the earliest, whereas for most of the harder cases it still has not nearly reached optimal performance at 2M timesteps. We again note that our approach shows greatly improved sample efficiency.

**Environments: Bilateral Trade on Atari 2600**   As a second, significantly higher-dimensional and challenging domain, we present a bilateral trade scenario on a modified Atari 2600 game (which are a state-of-the-art benchmark domain in single-agent RL). We use a two-player version of the game "Space Invaders", and introduce an artificial resource constraint: Each agent can only fire in the game if they have a bullet available. Initially, neither player has any bullets available. Throughout the episode, we give bullets to player 1, one at a time at stochastic intervals. Player 1 can then choose to offer the sell this bullet to player 2 by offering them a price, or Player 1 can choose to use the bullet themselves. Player 2 in turn can choose to accept or reject a particular offer at a particular price. If a trade takes place, the sales price is added to player 1's reward, and subtracted from player 2's reward. Additionally, we introduce a reward scale imbalance: Each time player 1 successfully shoots an alien invader, they get a reward of 0.1. However each time player 2 shoots an alien, they get a much higher reward of 1.0. Noting that even well-trained AI agents do not hit every single shot they take, we should still expect that player 2 be able to generate just under 1.0 reward from each bullet they fire, and player 1 a much smaller reward of just under 0.1.

Clearly there is more total reward generated if player 1 sells all their bullets to player 2, with the difference referred to as the "gains from trade" in economics. However, notice that this is not a mechanism design setting (there is no mechanism), and also that there are two Stackelberg equilibria: If player 1 is the leader, then their optimal strategy is to offer bullets to player 2 at just under player 2's average utility per bullet. Player 2 will best respond by accepting the trade, still generate small positive reward, and player 1 will receive almost the entirety of the gains from trade. In the second Stackelberg equilibrium, player 2 is the leader. Player 2's optimal strategy is to refuse any price higher than just above player 1's average utility per bullet; and player 1's best response is to offer to sell at that (low) price. In this scenario, player 1 will be left with little more reward than had they kept and used the bullets themselves, and player 2 will receive almost all the gains from trade.

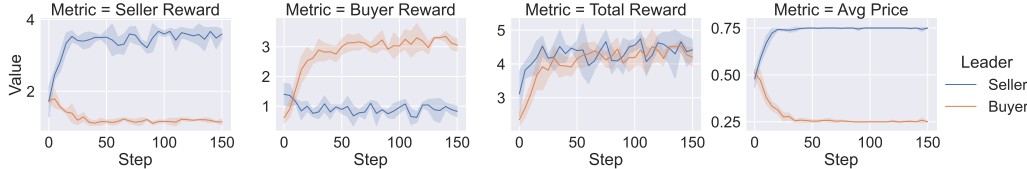

Figure 2: Performance of PPO+Meta-RL on Atari 2600 bilateral trade scenario. Plots show two Stackelberg equilibria: Agent 1 (seller) as leader (blue curves) and agent 2 (buyer) as leader (orange).

Figure 2 shows that our Meta-RL algorithm is able to successfully learn this for both equilibria. In this experiment we use discrete prices $(0, 0.25, 0.5, 0.75.1.0)$ for compatibility with the discrete Atari environment, so the results shown are the exact optimum.

## 6  CONCLUSION AND DISCUSSION

**Conclusion:** We have introduced a general framework for using multi-agent RL approaches to find Stackelberg equilibria in Markov games, and discussed how this encompasses several approaches in the literature while also conveying a much larger design space. As a second key contribution, we have proposed and evaluated a novel approach to Stackelberg learning that uses Meta-RL to implement the follower oracle. This shows the power of our main theorem, which enables this approach, but is also a key contribution itself. Our approach matches or exceeds performance to previous approaches at greatly reduced sample complexity in experiments. It enables Stackelberg learning in novel domains beyond the reach of previous approaches, which we show on a novel Atari 2600-based bilateral trade scenario. Finally, we show theoretically and experimentally the limits of Theorem 1, and in particular that RL algorithms can provably be unable to learn without the query-oracle special case construction.

**Discussion:** In addition to the technical results, we would like to offer a more high-level interpretation of the framework. A useful way to think about learning Stackelberg equilibria in Markov games is that they are, in a way, two problems in one: One, how does my strategy, i.e., choice of policy, affect the best-response of other agents? Two, how does my interactions with the environment, i.e., actions at each step, affect the reward I (and others) get? These are two very different problems, even operating at different levels—entire policy, versus action at each step. In a way, Theorem 1 is giving ways to reconcile the best-response "meta-level" and the environment-interaction "RL problem." In the general case, using techniques such as direct gradient descent or evolutionary policies, we focus on the best-response meta-level and either ignore the environment interaction (in evolutionary strategies) or subsume them inside an end-to-end differentiation (in direct policy gradient). In contrast, in the query-oracle special case, we focus on the environment interaction RL problem, and implicitly work the follower best-response into this. One way of looking at the contextual-policy follower oracle is that it makes the latter more feasible, by greatly reducing the number of leader queries compared to real environment interaction.

We hope that this Meta-RL for Stackelberg learning work will enable Stackelberg RL approaches to scale up to richer settings, both through the explicit, contextual-policy approach taken in this paper, as well as approaches that infer context through recurrent networks. Beyond this, we hope the framework of Theorem 1 will inspire novel ways of thinking about Stackelberg RL. One potential avenue for future work that we are excited about is to study approaches that explicitly take into account both the "meta-level" and "environment-interaction" problems outlined above. We believe that doing so could enable Stackelberg RL to scale to much more complex scenarios and open novel applications to it. If successful, this may enable automated learning of system design beyond traditional security games and mechanism design.

## REPRODUCIBILITY STATEMENT

All source code used for experiments will be submitted as part of the supplementary material, along with detailed instructions on how to recreate the experiments presented in this paper. Hyperparameters used are also listed in the appendix. We plan to release the source code of our experiments under an open-source license upon acceptance.

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

# A   PROOF OF THEOREM 1

We include here the proof of Theorem 1.

**Theorem 1.** *Given a Markov Game $\mathcal{M}$ and a follower equilibrium oracle $\mathcal{E}$, then:*

***[General Case]*** *If*

1. *the leader locally aims to solve a single-agent optimization problem $\mathcal{L}(\pi_L)$, and*

2. *for each choice of leader policy $\pi_L$, $\mathcal{L}$ computes the follower best-response $\mathcal{E}(\pi_L)$, and*

3. *$\mathcal{L}(\pi_L)$ evaluates the leader policy $\pi_L$ against the follower best-response $\mathcal{E}(\pi_L)$ in $\mathcal{M}$, i.e. the value of $\mathcal{L}(\pi_L)$ is $r_L(\pi_L, \mathcal{E}(\pi_L))$ in $\mathcal{M}$,*

*then an optimal solution $\pi_L^*$ to $\mathcal{L}$ together with the follower best-response $\mathcal{E}(\pi_L^*)$ form a Stackelberg equilibrium in $\mathcal{M}$.*

***[Query-Oracle Special Case]*** *Furthermore, if additionally*

4. *the follower oracle implementation $\mathcal{E}$ only requires query access to $\pi_L$, i.e. values $\pi_L(o)$ for one or more observations $o \in \mathcal{O}_L$ from $\mathcal{M}$, and*

5. *leader policies $\pi_L$ are invariant, i.e. acting the same during queries as they would in $\mathcal{M}$,*

*then $\mathcal{L}$ can be constructed as a POMDP.*

*Proof.* We show the proof in two parts, first for the general case, then the query-oracle special case.

**[General Case]** We first show the general case:
Assume $s_L^*$ optimally solves $\mathcal{L}$, i.e. $r_{L,\mathcal{L}}(s_L^*) = \max r_{L,\mathcal{L}}(\pi_L)$.

By condition 3, the leader's reward in $\mathcal{L}$ is the same as that in $\mathcal{M}$ when the followers play their best-response equilibrium, i.e. $r_{L,\mathcal{L}}(s_L^*) = \max r_{L,\mathcal{L}}(\pi_L) = \max r_{L,\mathcal{M}}(\pi_L, \mathcal{E}(\pi_L))$. This immediately means that $s_L^*$ together with $\mathcal{E}(\pi_L)$ form a Stackelberg equilibrium in $\mathcal{M}$. Condition 2 is only required implicitly to ensure that followers are playing their best-response equilibrium when the leader strategy $\pi_L$ is evaluated in $\mathcal{M}$. This shows the general case.

**[Query-Oracle Special Case]**

**Construction of $\mathcal{L}$:** Given a Markov Game $\mathcal{M}$ and a follower best-response oracle $\mathcal{E}$ that only requires query access to the leader strategy $\pi_L$, define *a new leader POMDP $\mathcal{L}$* as follows:

- **[initial segment]** For an initial number of steps in $\mathcal{L}$, each step performs one query from the follower oracle $\mathcal{E}$: If a given query wishes to determine the leader policy's response to observation $o$, then the leader will receive $o$ as its observation in $\mathcal{L}$, and the leader's action will be given to $\mathcal{E}$ as the response to its query. The leader will receive no reward in these steps.

- **[final segment]** Once a follower equilibrium $\pi_F$ has been determined, the remainder of $\mathcal{L}$ will be constructed from the original Markov game $\mathcal{M}$: We let followers act according to the computed follower equilibrium $\pi_F$ and treat them (including all their internal state) as part of the environment.

We now show that $\mathcal{L}$ is a POMDP.

**POMDP, setup:** Let the state of $\mathcal{L}$ be $z_t = (z_{\mathcal{E},t}, z_{\mathcal{M},t}, z_{F,t})$, the internal state of the follower equilibrium oracle (in the initial part of the $\mathcal{L}$), the state of the original Markov Game, and the internal state of the follower agents (in the final part of the $\mathcal{L}$). In the initial part wlog assume this is $(z_{\mathcal{E}}, 0, 0)$, and in the final part $(0, z_{\mathcal{M},t}, z_{F,t})$.

**POMDP, part 1:** By assumption, $\mathcal{E}$ only requires query access to $\pi_L$, i.e. if at timestep $t$, the oracle's internal state is $z_{\mathcal{E},t}$ and the oracle issues the query $o_t$, then the oracle's next internal state $z_{\mathcal{E},t+1}$ is a function of only $z_{\mathcal{E},t}$ and $q_t$, the leader's response to the query $o_t$. By the construction of

the first part of $\mathcal{L}$, we have that the leader's observation at timestep $t$ is precisely the oracle query $o_t$, and so it's action $a_t$ gives the oracle response $q_t$. Together, we get that the $\mathcal{L}$ state at time $t + 1$, $z_{t+1}$, is a function of only $z_t$ and $a_t$, showing that $\mathcal{L}$ is a POMDP in the initial part.

**POMDP, part 2:** In the final part of $\mathcal{L}$, at step $t$, the Markov Game state is $z_{\mathcal{M},t}$, and leader and follower observations depend only on this state, i.e. $o_{L,t} = o_{L,t}(z_{\mathcal{M},t})$ and $o_{F,t} = o_{F,t}(z_{\mathcal{M},t})$. In turn, both the follower actions $a_{F,t}$ as well as the next follower state $z_{F,t+1}$, only depend on $o_{F,t}$ and the current follower state $z_{F,t}$; therefore both depend only on $z_{\mathcal{M},t}$ and $z_{F,t}$. In turn, the next state of $\mathcal{M}$, $z_{\mathcal{M},t+1}$, depends on leader and follower actions, and therefore only on leader action, $z_{\mathcal{M},t}$ and $z_{F,t}$. Together, it follows that $z_{t+1}$ only depends on $z_t$ and the leader's action $a_{L,t}$, meaning the final part of $\mathcal{L}$ is Markovian.

We have therefore shown that $\mathcal{L}$ as a whole is a POMDP. We now show that an optimal policy in $\mathcal{L}$ forms a Stackelberg equilibrium.

**Stackelberg:** By the assumption that the leader policy is invariant, we have that if $\pi_L(o) = a$ in response to an oracle query, then $\pi_L(o) = a$ in the Markov Game $\mathcal{M}$ as well. Therefore, the follower equilibrium $\pi_F$ computed by the oracle at the end of the initial part of $\mathcal{L}$ is a best-response equilibrium to the strategy the leader plays in $\mathcal{M}$ in the final part of $\mathcal{L}$.

Now, by construction of $\mathcal{L}$, the leader reward given any $\pi_L$ in $\mathcal{L}$ is the same as the leader reward in $\mathcal{M}$ when followers play $\pi_F$, and by the above $\pi_F$ is indeed a best-response equilibrium, i.e. $r_{L,\mathcal{L}}(\pi_L) = r_{L,\mathcal{M}}(\pi_L, \pi_F) = r_{L,\mathcal{M}}(\pi_L, \mathcal{E}(\pi_L))$. Finally, by optimality of $\pi_L^*$ in $\mathcal{L}$, $\pi_L^* \in \mathrm{argmax}(r_{L,\mathcal{L}}(\pi_L))$, and therefore $\pi_L^* \in \mathrm{argmax}(r_{L,\mathcal{M}}(\pi_L, \mathcal{E}(\pi_L)))$. But this precisely means that $\pi_L^*$ and $\mathcal{E}(\pi_L^*)$ form a Stackelberg equilibrium in $\mathcal{M}$. $\qquad\square$

## B    THEOREM 1: LEADER MEMORY AND LEADER INVARIANCE

**Leader Memory.**   We state the query-oracle case of Theorem 1 for memory-less leader policies, i.e. leader policies that map directly from observations to actions. This is without loss of generality because for leader policies that use memory we may take the view that the leader policy operates on belief states, mapping belief state to action. In this view, the theorem applies as-is, and we query the leader policy on belief states. This would work well, for instance, if leader memory was implemented through a sufficient statistic. Alternatively, if we want to treat memory as intrinsic to the leader policy, queries become sequences of observations. In this view, the proof applies *mutatis mutandis*. The main technicality in this case is to reset internal state of the leader policy between queries, so that queries are well-defined. This is also important in order to ensure the leader invariance conditions (an unrestricted LSTM could easily allow a leader to distinguish queries from real game).

**Leader Invariance.**   It may be possible to give the leader policy memory beyond the two cases above, i.e. memory with state that carries through between follower oracle queries and/or to real play. In any such cases, it is necessary that the leader policy be *invariant*, meaning it is acting the same during the initial segment (i.e. oracle queries) and the final segment (i.e. original game) of the constructed leader POMDP $\mathcal{L}$. This has not been stated explicitly in previous works, but is a critical part of ensuring convergence to the correct equilibrium. If the leader policy were to act differently during the oracle queries, it could "trick" followers into suboptimal behavior that gives the leader better reward but is not a best-response, and thus not a Stackelberg equilibrium. For instance, in an iterated prisoners dilemma, a leader could pretend to be playing tit-for-tat during oracle queries, leading followers to cooperate; and could then defect during the actual game. We show this experimentally in Appendix C.2. Invariance is easily ensured if the leader policy cannot distinguish queries and real play, which is generally true for memory-less policies. Alternatively the leader policy could be explicitly constrained to be invariant, e.g. through an appropriate loss term.

## C    LIMITATIONS OF THEOREM 1

We now present experimental evidence of the limitations of our main Theorem. In particular, we will show that violating any of the conditions of the theorem can lead to learning failure.

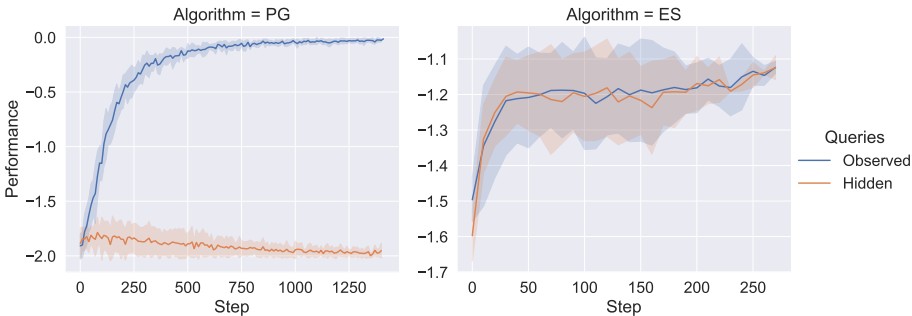

Figure 3: Performance of contextual-policy approach with hidden and rolled-out oracle queries in iterated prisoners dilemma.

## C.1 NON-POMDP

An interesting question we asked earlier is whether rolling out the follower queries into the leader episode to form a POMDP is strictly necessary. In the next section we will present a proof that this is the case, but we also show this experimentally here. Figure 3 shows the performance of our approach on a slightly modified iterated prisoner's dilemma (see Appendix D for full payoff matrices). We show our standard setting where queries are part of the leader episode, as well as a setting where they are hidden from the leader. The hidden-queries setting fails to learn a sensible behavior. This is consistent across learning rates, and across algorithms. Note however that this only applies to RL algorithms. An approach that operates directly on the policy space such as Evolutionary Strategies is still able to learn successfully, as shown on the right hand side of Figure 3.

## C.2 NON-INVARIANT LEADER

We also experimentally illustrate the leader invariance condition in Theorem 1 in the iterated prisoner's dilemma setting. For simplicity, we emulate memory for the leader policy by concatenating a binary variable to its observation, set to 0 during the first five steps of each episode (the queries), and 1 afterwards.[2] As can be seen in Figure 4, when given access to this additional variable, the leader gains significantly higher reward. The leader policy effectively learns to act as if it was playing tit-for-tat during the queries, thereby inducing the follower to respond by cooperating; the leader then always defects during the actual game, thereby achieving maximum reward. This is not a Stackelberg equilibrium.

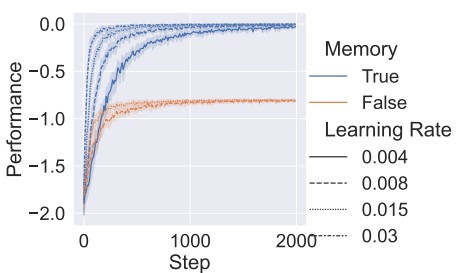

Figure 4: Leader reward with invariant and non-invariant policies in iterated prisoners dilemma.

## C.3 LEADER REWARD DURING FOLLOWER LEARNING

One condition of Theorem 1 is that the leader only be evaluated against followers who are best-responding. If the follower oracle is implemented using learning dynamics observable to the leader, this means that the leader must not receive reward during this learning phase. If the leader did receive reward, this could give the leader the wrong optimization target. Imagine for instance a setting where the leader has one strategy choice corresponding to a quickly-learnable follower best-response strategy that gives medium reward to the leader, and another leader strategy choice corresponding

---

[2]A neural network could learn to extract the same discriminator from a step counter, and a recurrent network could easily learn to keep such a counter.

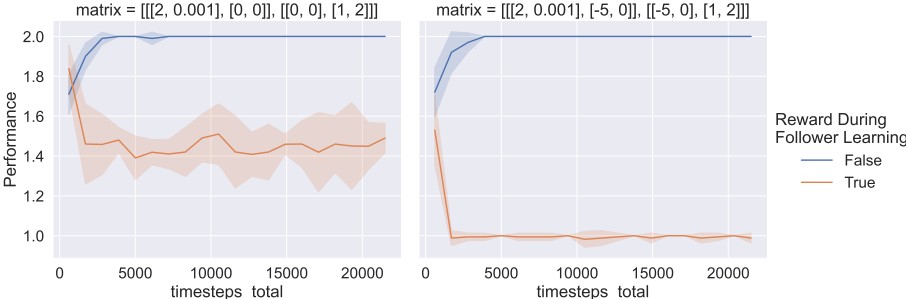

Figure 5: Leader performance with and without reward during follower Q-learning. Clearly the leader fails to learn the sole Stackelberg equilibrium (reward 2.0) if reward is given during follower learning. Plots show reward during actual play only, i.e. without reward during follower Q-learning, as this is the relevant quantity for Stackelberg equilibria.

to a slow-to-learn follower strategy with high leader reward. We can easily simulate this using a slightly modified version of the "Battle of the Sexes" single-shot matrix game we used as an example in the introduction. In this, we modify the follower reward so that the leader-preferred option gives the follower very little reward. We then couple this with carefully chosen (but entirely reasonable) Q-learning hyperparameters for the follower. As a result, a leader who receives reward during the follower learning phase is not able to reliable learn the correct equilibrium anymore, even in such a simple game, as Figure 5 shows. If we further modify the game to penalize the leader for coordination failure, this can even lead to the leader consistently learning the wrong coordination choice, as the right-hand plot shows.

Notice however that this (reward throughout follower learning) is also a valid target to optimize for, where the leader aims to optimize its expected return taking into account that followers may need some time to adjust to the leader's behavior. In the case of Balaguer et al. (2022) this is the intent, especially with regards to designing mechanisms for human participants as followers.

## C.4 Continuous Follower Learning

Finally, virtually all previous approaches in the literature use some sort of learning dynamics to implement the follower oracle. A tempting way of improving learning speed in such a paradigm would be to retain follower policies between leader updates. That is, if at the end of the leader learning iteration $t$, the follower is best responding using strategy / policy parameters $\phi_{t,\text{end}}$, then instead of initializing follower weights $\phi_{t+1,\text{start}}$ randomly, set $\phi_{t+1,\text{start}} = \phi_{t,\text{end}}$. Under the assumption that the leader policy only changed a little, and the conjecture that therefore the optimal follower policy only changed a little, this should allow follower learning to start from very near the optimum, and thus hopefully require much short inner (follower learning) loops. However, this has some drawbacks. For one, it makes the

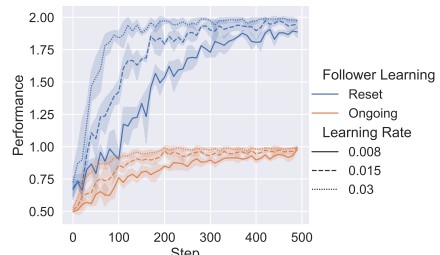

Figure 6: Leader reward with a Q-learning follower on Battle of the Sexes, where the follower initializes a blank Q-table each episode (blue) or keeps their previous Q-table (orange).

leader learning problem non-stationary. Beyond this, it can lead to learning failure, if both leader and follower get stuck on a local optimum. Figure 6 shows this in practice on the "Battle of the Sexes" example, where non-resetting follower learning can lead to convergence to the follower-preferred choice rather than the Stackelberg equilibrium.

## D    PROOF OF NON-POMDP DIVERGENCE

We now present a theorem that shows that the query-oracle special case is meaningful. A priori it is not clear that the query-oracle POMDP construction is strictly necessary, or if standard RL algorithm could also learn without it. Without the POMDP construction, the leader would effectively always play against followers who immediately best respond. The following theorem shows that this cannot work.

**Theorem 2.** *There exists a Stochastic Markov Game $\mathcal{M}$ where neither tabular Q-learning nor policy gradient can learn the optimal policy for the leader, if the follower agent always best-responds (without the construction of the query-oracle special case of Theorem 1).*

*Proof.* We consider a slight variation of the iterated prisoner's dilemma discussed in the main text. Consider payoff matrices $L = \begin{pmatrix} 0 & -2 \\ -1 & -3 \end{pmatrix}$ and $F = \begin{pmatrix} -1 & 0 \\ -3 & -2 \end{pmatrix}$ denoting the payoff to the leader and follower agent respectively. The leader chooses the row, the first row denoting "cooperate" or $C$ and the second row "defect" or $D$, and similarly the follower chooses the column. Notice that these are the standard prisoner's dilemma payoff matrices, except the top and bottom row for the leader have been switched.

Let each agent's observation space be a one-step memory of the *other* agent's previous action, that is, there are three possible observations $o_0$, $o_C$ and $o_D$. At the first step of each episode, both agents observe $o_0$. If at step $t$ the leader cooperates and the follower defects, then at step $t + 1$, the leader observes $o_D$ ("other agent defected") and the follower observes $o_C$ ("other agent cooperated"). We also write $s_{CD}$ for this state if we want to refer to both agents. In particular $s_{CD}$ corresponds to leader reward -2 for the leader and 0 for the follower (top right corner of both matrices). This is a simplification of the setting presented in the main text, but without loss of generality in the case of iterated prisoner's dilemma, and also defines a valid stochastic Markov game in its own right. Let us define an episode of our SMG to be $h$ iterations of the matrix game, where $h$ denotes the horizon or episode length of the game. As a preliminary, notice that for any leader policy, the follower best-response is always deterministic. This is easy to check.

It is also easy to see that the optimal leader policy is to cooperate on the first step, and to then play tit-for-tat. That is, if the follower cooperated, the leader cooperates in the following step. If the follower defects, the leader defects in return. If the leader plays this policy, then the follower will in turn always cooperate, leading to leader episode reward 0, clearly the optimum. Using the construction in the query-oracle special case of Theorem 1, this optimal leader policy can be learned using standard RL algorithms. What we will now show is that if the follower best-responds without that construction, i.e. immediately without queries folded into the leader sample batch, then standard RL algorithms will diverge. This is independent of choice of hyperparameters, but as a matter of principle.

Intuitively, the problem is one of a missing counterfactual: Notice that for the leader tit-for-tat Stackelberg equilibrium, it is essential that the leader commits to defecting if the follower defects. But notice also that when the leader plays this tit-for-tat policy, the follower will always best respond, and so the leader will never actually see a follower defection. But this also means that it cannot accumulate a gradient for this hypothetical behavior.

We now make this formal. Consider first the case of (tabular) Q-learning. Let $q(s, a)$ be the (leader's) Q-value of taking action $a$ in state $s$. We let $\alpha$ denote the learning rate and $\gamma$ the discount factor. Given an experience $(s, a, r, s')$ we update Q-values as follows:

$$q(s, a) \quad \leftarrow \quad (1 - \alpha) \cdot q(s, a) \quad + \quad \alpha \cdot \big(r + \gamma \max q(s', .)\big) \tag{2}$$

As a convenient shorthand and as a slight abuse of notation, we will define $\theta$ as follows:

$$\theta_s = \begin{cases} 0 & \text{if } q(s, D) \leq q(s, C) \\ 1 & \text{if } q(s, D) > q(s, C) \end{cases} \tag{3}$$

In words, we let $\theta_s = 1$ denote that the current leader policy given the $q(s, a)$ values will defect in state $s$, and 0 if the leader will cooperate in state $s$. We can then write $\theta = (\theta_0, \theta_C, \theta_D)$ for the entire leader policy induced by the current Q-table. $\theta = (0, 0, 0)$ would denote a leader policy that always

cooperates, $\theta = (1, 1, 1)$ denotes a leader always defecting, and $\theta = (0, 0, 1)$ denotes the (optimal) tit-for-tat strategy.

Now consider the case of tabular Q-learning with parameter noise exploration. In this, we collect experiences from any of the eight possible deterministic leader policies. Note also that the leader action on the initial step does not affect the follower's best response strategy; and it does not influence Q-table updates for the non-initial observations $o_C, o_D$ (because $o_0$ will never be revisited and so the reward generated from $o_0$ can never appear in a Q-table update or indeed in a reward-to-go calculation in a policy gradient algorithm). We can therefore disregard the leader's initial action and for brevity focus only on the four cases $\theta = (\star, 0, 0)$, $\theta = (\star, 0, 1)$, $\theta = (\star, 1, 0)$ and $\theta = (\star, 1, 1)$. It is easy to see that for $\theta = (\star, 0, 1)$ the follower best-response is to always cooperate, and for the other three cases it is to always defect. We may therefore encounter experiences of the following form:

$$
\begin{array}{lcl}
\theta = (\star, 0, 0) & \rightarrow & (o_D, C, -2, o_D) \\
\theta = (\star, 0, 1) & \rightarrow & (o_C, C, 0, o_C) \\
\theta = (\star, 1, 0) & \rightarrow & (o_D, C, -2, o_D) \\
\theta = (\star, 1, 1) & \rightarrow & (o_D, D, -3, o_D)
\end{array}
$$

It is easy to see that under usual Q-learning update rules and for any choice of learning rate, we will have that in the limit $q(o_D, C) = -2 \cdot g$ (lines 1, 3) and $q(o_D, D) = -3 \cdot g$ (line 4) where $g = \frac{1-\gamma^{h/2}}{1-\gamma}$ is a term from the discount factor $\gamma$. Crucially we have that $q(o_D, C) > q(o_D, D)$, and therefore the policy will converge toward $\theta = (\star, \star, 0)$, which is not optimal. This holds for any choice of learning rate, discount factor and exploration parameters (as any mix of the above trajectories will lead to this).

For the $\epsilon$-greedy case, let $\theta_s^\epsilon = \theta_s + (-1)^{\theta_s}(\epsilon/2)$. That is, if our current Q-table induces the deterministic policy $\theta$, then $\theta_s^\epsilon$ gives the probability of choosing action $D$ in state $s$ in the $\epsilon$-greedy case. It is easy to see that for sufficiently small $\epsilon$ and $\theta_s^\epsilon = (\star, \epsilon, 1 - \epsilon)$ the follower best-response is still to always cooperate, and for any other $\theta_s^\epsilon$ the follower best-response is to always defect. Therefore in particular, no matter which way a particular leader action is sampled, the follower will best-respond in the same way (only depending on the leader policy as a whole, not the particular leader action sampled). In turn this means that $q(o_D, C)$ can only continue to accumulate $-2$ terms, and $q(o_D, D)$ can only continue to accumulate $-3$ terms, and the policy will converge toward $\theta = (\star, \star, 0)$, which is not optimal.

To show this for policy gradient, let the leader policy be parametrized by $theta$ as above, i.e. let $\theta_o$ be the probability that the leader policy defects given observation $o$, and $1 - \theta_o$ the probability that the leader cooperates given $o$. Recall the basic REINFORCE gradient update rule:

$$
\theta \leftarrow \theta + \alpha G_t \nabla_\theta \ln \pi_\theta(a_t | o_t) \tag{4}
$$

Here $G_t$ denotes the (discounted) "reward to go", i.e. $G_t = r_t + \gamma r_{t+1} + \gamma^2 r_{t+2} \ldots$ as usual. A very similar argument as in the Q-learning case now holds to show that the reward-to-go from cooperating when observing $o_D$ will always be larger in expectation than the reward-to-go from defecting, because $r_t$ when defecting is smaller than $r_t$ when cooperating given $o_D$ and the remainder of the sum in $G_t$ is the same in expectation. This in turn pushes gradients toward cooperation, and away from the optimal tit-for-tat policy. □

The above holds for tabular Q-learning and basic policy-gradient with direct parametrization, but likely can be extended to further RL algorithms such as DQN or actor-critic.

Notice the key difference in the query-oracle POMDP construction: In this, the oracle must query the leader policy for its action given $o_D$ at least once in the initial "oracle" segment of the episode. That action therefore sees as its reward to go the reward from the entire final segment, i.e. the entire episode reward of the original Markov game. Intuitively, the leader gets to see at least one experience where it retaliates on a follower defection and this leading to an entire episode of cooperation and good rewards. Without the oracle query, the leader never gets to see this, and cannot learn from it. It may still see experiences where it retaliates for defection, but these will be from within the actual episode, will not influence follower behavior, and will lead to strictly worse rewards than

cooperating. Finally, it is also clear that this only applies to typical RL algorithms that learn on taking actions in individual steps. Approaches that learn on the policy space as a whole, such as evolutionary strategies, are not affected by this (as indeed they never look at individual steps and actions at all).

## E    NECESSARY CONDITIONS FOR STACKELBERG CONVERGENCE

It may also be interesting to consider the inverse direction of Theorem 1, i.e. what are necessary conditions that follow from Stackelberg convergence. The resulting theorem is not very strong, but still informative, as it suggests avenues for future research. Recall that in Theorem 1 we map a Markov game to a single-agent RL problem (POMDP) for the leader. In the general case this is simply taking the leader's view of the original Markov game as-is, and in the query-oracle special case we construct a POMDP that incorporates oracle queries. We then show that a solution to the leader's POMDP together with the follower best-response forms a Stackelberg Equilibrium.

Consider now the reverse: Suppose we are given some mapping from Markov game to leader POMDP, and a guarantee that no matter the original Markov game, an optimal solution to the leader POMDP it maps to forms part of a Stackelberg equilibrium. What needs to be true of any such mapping? We formulate this here in a slightly more general manner, in that we also allow an additional (not necessarily identity) mapping between leader policies in the Markov game and the POMDP.

**Theorem 3** (Necessity). *Suppose we are given mappings $\mathcal{L} : \mathcal{M} \mapsto \mathcal{L}(\mathcal{M})$ and $l : \Pi_{\mathcal{L}} \to \Pi_{\mathcal{M}}$. $\mathcal{L}$ maps any Markov Game $\mathcal{M}$ to a single-agent RL problem, and $l$ maps policies in $\mathcal{L}$ to policies in $\mathcal{M}$. Furthermore suppose that whenever a policy $\pi_{L,\mathcal{L}}$ optimally solves $\mathcal{L}(\mathcal{M})$, then $l(\pi_{L,\mathcal{L}})$ together with $\mathcal{E}(l(\pi_{L,\mathcal{L}}))$ are a Stackelberg equilibrium in $\mathcal{M}$. Then the following two conditions must be true of $\mathcal{L}$ and $l$.*

1. *The leader reward in $\mathcal{L}$ is maximized by the same choice of strategy as the leader reward in $\mathcal{M}$ when followers play $\mathcal{E}(\pi_L)$, i.e.*

$$l\big( \arg\max r_{L,\mathcal{L}}(\pi_L)\big) \subseteq \arg\max r_{L,\mathcal{M}}(\pi_L, \mathcal{E}(\pi_L))$$

2. *$\mathcal{L}$ implements a follower equilibrium oracle $\mathcal{E}(\pi_L)$*

*Proof (Sketch).* The first condition immediately follows from the problem statement. For the second condition, consider that given full freedom in choosing $\mathcal{M}$, we can construct $\mathcal{M}$ so as to let $\mathcal{E}$ be any arbitrary function from leader to follower policy space. Similarly, we can choose $\mathcal{M}$ so that $r_L$ is any arbitrary function. Both of these follow from cardinality arguments, and the observation that since Markov games may be partially observable we are essentially unrestricted in the complexity of the Markov game we choose to construct even for small strategy spaces. Since by the first condition $\mathcal{L}$ needs to compute $r \circ \mathcal{E}$, both of which can be arbitrary, it thus also needs to compute $\mathcal{E}$.    □

The main difference to the conditions in Theorem 1 is that we can only show that the argmax of the leader reward needs to be that of the original Markov game, not that the rewards need to be identical. This is in a way trivial (of course Theorem 1 still holds if we scaled leader rewards in the leader learning problem by a constant factor), but it also suggests that reward shaping may be a viable technique to accelerate leader learning, potentially still with provable Stackelberg equilibrium guarantees.

## F    FURTHER EXPERIMENT DETAILS: ITERATED MATRIX GAMES

**Environments**    We use the 12 canonical symmetric matrix games identified in Robinson & Goforth (2005) and also used by Balaguer et al. (2022). We construct Markov games from these matrices by concatenating multiple iterations into an episode, and giving both agents one-step memory of both agents' action in the previous step. We use $n = 10$ steps per episode. Table H shows the payoff matrices for all the Markov games, reported on the same scale as the figures. During training, we scale rewards to be centered at 0, i.e. taking values $-1.5, -0.5, 0.5, 1.5$, but we report results offset to match the reward scales used by Balaguer et al. (2022). This has no effect on comparability of results.

**Algorithm.** We focus on the contextual policy meta-learning approach described in subsection 4 for followers, and standard RL for the leader: At the beginning of each episode, the leader is queried (as part of the episode rollout) for its action in each possible state of the environment. Its responses are then concatenated to the follower observation. In a pre-training phase, we train the follower against randomly sampled leader policies. In the main training phase, we then train the leader against the follower meta-policy. Algorithm 1 in the Appendix details this in pseudo-code. An advantage of the generality of our framework is that it is agnostic to which specific RL algorithm is used. We generally use a standard policy gradient (PG) algorithm for the followers, although our results do not depend on this specific choice.

Algorithm 1 details the two-phase learning algorithm we use. In all the experiments shown in the main text, we use policy gradient to train the follower meta-policy in the pre-training loop. We use PG (Sutton et al., 1999), PPO (Schulman et al., 2017) and DQN (Mnih et al., 2013; 2015) in the main training loop, as indicated in the respective figures. We use linear models, and disable exploration in the leader policy while pre-training the follower and vice versa. Table H lists the hyperparameters used for each of these algorithms. Any hyperparameters not listed were left at default values in **rllib** version 2.0.0. All experiments were run with a single rollout worker (per experiment), and using Torch.

**Equilibrium Verification.** At the end of every experiment, we freeze the leader policy and further train the follower policy for $n = 50$ iterations. Unlike in the pre-training phase, we here train them only against the specific leader policy trained in the main training loop. This is to further verify that the policies indeed form a Stackelberg equilibrium, and in particular that the follower meta-policy is best-responding to the trained leader. If this is the case, we should not see any change in leader or follower performance in this post-training phase. If the follower meta-policy was *not* already best-responding to the leader, we may see an increase in follower performance during this post-training phase. In all of the experiments in this paper (except the ones designed to show failure modes) we see no follower improvement, i.e. behavior consistent with a Stackelberg equilibrium. This is not shown in the training curves in the figures, but can be reproduced from the source code.

**Implementation and Environment.** All experiments were implemented using Ray / RLlib 2.0.0 (Liang et al., 2018). Experiments were run on recent Intel Xeon processors with a single core and 2GB RAM per experiment.

**Hyperparameter Tuning.** Learning rates and batch sizes were tuned using grid search, with some additional tuning using the HyperOpt Python package (Bergstra et al., 2013), yielding no further improvement however.

## G  FURTHER EXPERIMENT DETAILS: ATARI 2600

**Environment.** We modify the Atari 2600 game "Space Invaders". We read from emulator RAM to detect when a shot has been fired, and by which player. Separately in a Python wrapper we keep a count of how many shots each player has available. We decrement this whenever we detect that the player fired a shot. If the Python variable keeping track of the available bullets reaches zero, we overwrite the player action that is fed to the Atari emulator to not-firing. Both players start with zero available bullets, but we increment the bullets available to player 1 at stochastic intervals for up to a total of five times per episode.

We implement a bilateral trade between agents: The selling agent may offer a price, and the buying agent may choose to accept this price.

**Neural Network Architecture.** This is implemented by augmenting both action and observation space, both providing a dictionary of both the underlying Atari action/observation, as well as the new economic action and observations.

The action space contains the original Atari action, as well as the trading action. For the seller, the trading action is picking one of several discrete price points, where we choice $n = 5$ price points ranging from 0 to 1 in 0.25-step increments. For the buyer, instead of giving a discrete buy / don't-buy action, we let the buyer policy set a maximum price it is willing to buy. If the offered sales

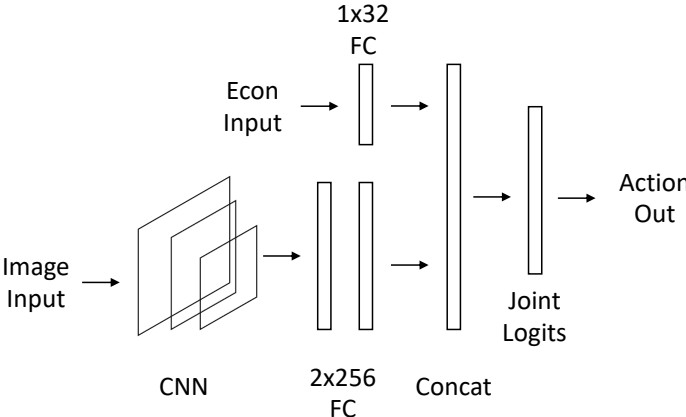

Figure 7: Neural Network Architecture used in the Atari 2600 bilateral trade experiments.

price is below the maximum buying price of the buyer, the trade happens, and the price paid is that set by the selling agent. It is easy to see that this is equivalent to letting the buying agent observe the price offer and respond with acceptance or rejection. We chose this implementation as it makes implementing the follower oracle easier when the buyer is Stackelberg leader, but it does not affect the outcome.

In the observation space, we provide a Dictionary to each agents containing both the original Atari 2600 image observation, as well as all the relevant economic information (number of bullets the agent currently has available, if applicable price offered by the other agent, whether a trade is current being proposed). In the neural network, we run these economic features through a separate fully connected layer, which feeds into a joint logits layer. The Atari input is run through default RLlib CNN and fully connected layers. Figure 7 shows this neural network architecture as a diagram.

**Algorithm.** We use standard PPO for both the leader and follower. Hyperparameters were taken from RLlib tuned examples and are listed at the end of Table H. For convenience, we initialize weights of the CNN and default RLlib FC layers to weights obtained from training agents in the unmodified game. This speeds up training, but is not strictly necessary. We utilize the same Meta-RL approach as we do in the iterated matrix game experiments: We first train a meta-follower. In this phase, we let the gameplay actions of the leader agent be controlled by an agent trained on the unmodified game, but we randomize the leader's economic actions. Once this meta-follower training has finished, we train the leader. In this phase, the meta-follower weights are frozen, and only the leader policy is trained. In the Atari experiments, we let the meta-follower query the leader immediately before each trade rather than at the start of the episode, as this allows us to fold the queries into the trading exchange.

## H   FURTHER DETAILS ON PERFORMANCE COMPARISONS

In Figure 1 we compare our Meta-RL approach with the PPO+Q-learn approach of Brero et al. (2022) and the ES-MD approach of Balaguer et al. (2022).

For Brero et al. (2021a), we implement follower Q-learning using information therein. Hyperparameters for both the leader and the follower were tuned using the HyperOpt package (Bergstra et al., 2013). In Figure 1 we plot learning curves up to 200k timesteps, as our approach converges before that point. We show in Figure 8 learning curves until 2M timesteps. We can see that in some cases PPO+Q-learn eventually converges to the optimum, while in the majority of cases this still has not happened by 2M timesteps.

For Balaguer et al. (2022), we estimate their performance from Figure 2 therein. Notice that that figure is *not* a learning curve, but represents a single inner loop at the end of their training procedure.

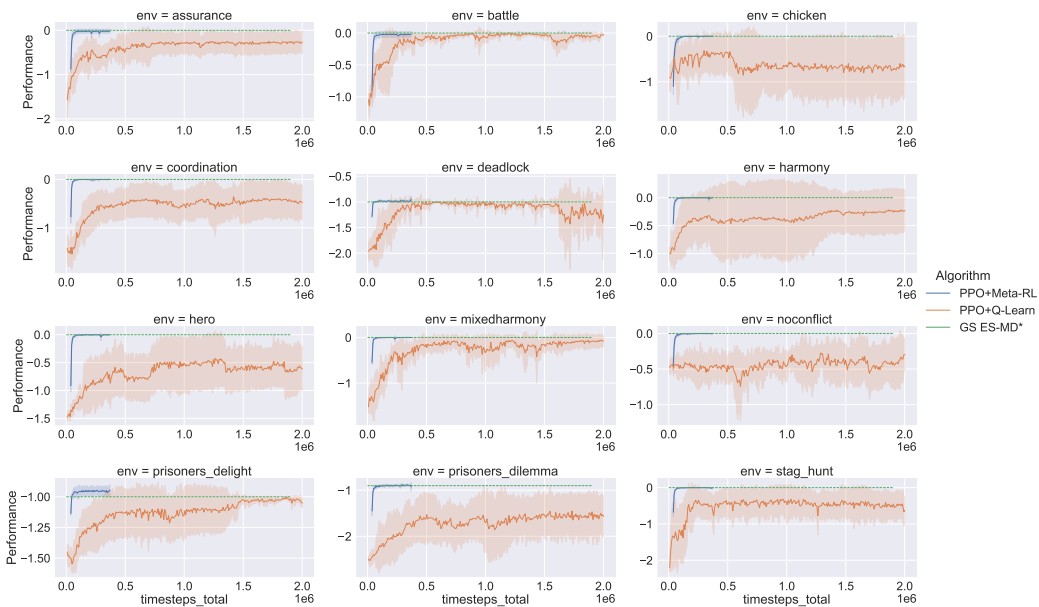

Figure 8: Performance on symmetric matrix games (see Figure 1) up to 2M timesteps.

In the ES-MD case, Balaguer et al. (2022) report their performance after 1.28 billion environment steps. In the Diff-MD case, a comparison of sample complexity is difficult, as that approach uses a description of the environment rather than sample access. The closest we can come to a like-for-like comparison is noting that Balaguer et al. (2022) report performance for Diff-MD after 500k computed expected episode returns with 10-step episodes. In some sense this could be seen to be equivalent to 5M environment steps as a lower bound.

---

**Algorithm 1** Contextual Policy

---

**Pre-Training Loop**
Initialize follower policy $\pi_F$
**for** each pre-training iteration **do**
 **for** each episode per sample batch **do**
  Sample a random leader policy $\pi_L^r$
  **for** each $o_L \in O_L$ **do**
   Query $\pi_L^r$ for $\pi_L^r(o_L)$
  **end for**
  Set context $\omega = \pi_L^r(o_L), o_L \in O_L$
  **for** each episode step **do**
   Return $o_{L,t}$ to leader, $(\omega, o_{F,t})$ to follower
   Step environment using $a_{L,t} = \pi_L^r(o_L), a_{F,t} = \pi_F(\omega, o_{F,t})$
  **end for**
 **end for**
 Update follower policy $\pi_F$ using collected sample batch using PG/PPO/DQN
**end for**
**Main Training Loop**
Initialize leader policy $\pi_L$
**for** each training iteration **do**
 **for** each episode per sample batch **do**
  **for** each $o_L \in O_L$ **do**
   Query $\pi_L$ for $\pi_L(o_L)$
  **end for**
  Set context $\omega = \pi_L(o_L), o_L \in O_L$
  **for** each episode step **do**
   Return $o_{L,t}$ to leader, $(\omega, o_{F,t})$ to follower
   Step environment using $a_{L,t} = \pi_L^r(o_L), a_{F,t} = \pi_F(\omega, o_{F,t})$
  **end for**
 **end for**
 Update leader policy $\pi_L$ using collected sample batch using PG/PPO/DQN
**end for**

---

Table 3: Payoff Matrices used in the matrix-game experiments

**Iterated Matrix Games (Figure 1 etc.)**

| Name | Leader Payoff | Follower Payoff |
|---|---|---|
| prisoners dilemma | $\begin{pmatrix} -1 & -3 \\ 0 & -2 \end{pmatrix}$ | $\begin{pmatrix} -1 & 0 \\ -3 & -2 \end{pmatrix}$ |
| stag hunt | $\begin{pmatrix} 0 & -3 \\ -1 & -2 \end{pmatrix}$ | $\begin{pmatrix} 0 & -1 \\ -3 & -2 \end{pmatrix}$ |
| assurance | $\begin{pmatrix} 0 & -3 \\ -2 & -1 \end{pmatrix}$ | $\begin{pmatrix} 0 & -2 \\ -3 & -1 \end{pmatrix}$ |
| coordination | $\begin{pmatrix} 0 & -2 \\ -3 & -1 \end{pmatrix}$ | $\begin{pmatrix} 0 & -3 \\ -2 & -1 \end{pmatrix}$ |
| mixedharmony | $\begin{pmatrix} 0 & -1 \\ -3 & -2 \end{pmatrix}$ | $\begin{pmatrix} 0 & -3 \\ -1 & -2 \end{pmatrix}$ |
| harmony | $\begin{pmatrix} 0 & -1 \\ -2 & -3 \end{pmatrix}$ | $\begin{pmatrix} 0 & -2 \\ -1 & -3 \end{pmatrix}$ |
| noconflict | $\begin{pmatrix} 0 & -2 \\ -1 & -3 \end{pmatrix}$ | $\begin{pmatrix} 0 & -1 \\ -2 & -3 \end{pmatrix}$ |
| deadlock | $\begin{pmatrix} -2 & -3 \\ 0 & -1 \end{pmatrix}$ | $\begin{pmatrix} -2 & 0 \\ -3 & -1 \end{pmatrix}$ |
| prisoners delight | $\begin{pmatrix} -3 & -2 \\ 0 & -1 \end{pmatrix}$ | $\begin{pmatrix} -3 & 0 \\ -2 & -1 \end{pmatrix}$ |
| hero | $\begin{pmatrix} -3 & -1 \\ 0 & -2 \end{pmatrix}$ | $\begin{pmatrix} -3 & 0 \\ -1 & -2 \end{pmatrix}$ |
| battle | $\begin{pmatrix} -2 & -1 \\ 0 & -3 \end{pmatrix}$ | $\begin{pmatrix} -2 & 0 \\ -1 & -3 \end{pmatrix}$ |
| chicken | $\begin{pmatrix} -1 & -2 \\ 0 & -3 \end{pmatrix}$ | $\begin{pmatrix} -1 & 0 \\ -2 & -3 \end{pmatrix}$ |

**Single-Shot Matrix Game (Appendix C)**

| | | |
|---|---|---|
| battle of the sexes | $\begin{pmatrix} 2 & 0 \\ 0 & 1 \end{pmatrix}$ | $\begin{pmatrix} 1 & 0 \\ 0 & 2 \end{pmatrix}$ |

**Modified Prisoner's Dilemma (Theorem 2)**

| | | |
|---|---|---|
| prisoners dilemma modified | $\begin{pmatrix} 0 & -2 \\ -1 & -3 \end{pmatrix}$ | $\begin{pmatrix} -1 & 0 \\ -3 & -2 \end{pmatrix}$ |

Table 4: Hyper-Parameter Configuration Table

**Follower Policy Gradient**

| Hyper-Parameter | Value | Hyper-Parameter | Value |
|---|---|---|---|
| algorithm | PG | rollout_fragment_length | 100 |
| lr | 0.02 | train_batch_size | 100 |
| iterations | 500 | batch_mode | complete_episodes |

**Leader Policy Gradient**

| Hyper-Parameter | Value | Hyper-Parameter | Value |
|---|---|---|---|
| algorithm | PG | rollout_fragment_length | 100 |
| lr | 0.156 | train_batch_size | 100 |
| iterations | 1200 | batch_mode | complete_episodes |

**Leader PPO**

| Hyper-Parameter | Value | Hyper-Parameter | Value |
|---|---|---|---|
| algorithm | PPO | rollout_fragment_length | 1000 |
| lr | 0.008 | train_batch_size | 1000 |
| entropy_coeff | 0.0 | sgd_minibatch_size | 1000 |
| iterations | 500 | batch_mode | complete_episodes |

**Leader DQN**

| Hyper-Parameter | Value | Hyper-Parameter | Value |
|---|---|---|---|
| algorithm | SimpleQ | rollout_fragment_length | 10 |
| lr | 0.001 | train_batch_size | 1024 |
| learning_starts | 5000 | exploration_type | ParameterNoise |
| exploration_initial_stddev | 1.0 | exploration_random_timesteps | 0 |
| iterations | 20000 | batch_mode | complete_episodes |

**Atari PPO**

| Hyper-Parameter | Value | Hyper-Parameter | Value |
|---|---|---|---|
| train_batch_size | 5000 | rollout_fragment_length | 100 |
| sgd_minibatch_size | 100 | num_sgd_iter | 10 |
| lambda | 0.95 | kl_coeff | 0.5 |
| clip_param | 0.1 | vf_clip_param | 10.0 |
| entropy_coeff | 0.01 | lr | 0.001 |
| num_rollout_workers | 10 | num_envs_per_worker | 5 |

