# OpenReview forum: "Oracles and Followers: Stackelberg Equilibria in Deep Multi-Agent Reinforcement Learning"
_ICLR.cc/2023/Conference — Submitted to ICLR 2023_

### Official Review · Reviewer_3yUN · 2022-10-24

**Confidence:** 3
**Correctness:** 4
**Technical Novelty And Significance:** 3
**Empirical Novelty And Significance:** 2
**Recommendation:** 8

**Clarity, Quality, Novelty And Reproducibility:**

Clarity: the work is well-written and easy to follow.

Novelty: in some sense, the work is novel, as it presents an original viewpoint, but it also is just building on/generalizing previous work e.g. the cited Brero et al. paper.

Reproducibility: reproducibility seems fine, the code included looks after looking through it briefly.

Quality: the work is high quality although the main theorem and experiments, while they are enough to support the argument, are not too impressive.

**Strength And Weaknesses:**

Strengths: The paper presents a very clear paradigm which unifies many existing approaches. This is an important topic, as shown by the cited prior work that involves using RL-style approaches in Stackelberg games, but much prior work has somehow lacked the clarity of perspective which this paper provides.

Weaknesses: the main theorem is not very exciting in some sense. And the experiments are not particularly impressive (they mainly look at very toy matrix games, when there are many more interesting problems that in principle can be described by this framework).

**Summary Of The Paper:**

The paper deals with the question of learning Stackelberg equilibria -- games with a leader-follower structure -- an important topic with many relevant applications. The main contribution is to show that, given access to a best-response oracle for the follower, this problem can be reduced to learning in a POMDP consisting of two phases. In some simple games, the authors then implement this reduction experimentally. A crucial requirement is that in the first phase of the POMDP, the leader does not adapt their policy; the authors also experimentally test this point.

**Summary Of The Review:**

I recommend weak acceptance simply because in the end, I was glad to read this paper and think it is a positive contribution to the community.

Update after reading author's rebuttal: due to the presence of new experiments (and clarification of comparison to previous methods) which are indeed more convincing, I've increased my score.

---

> ### Author Response · Authors · 2022-11-18
> **Response to reviewer 3yUN**
>
> Dear Reviewer 3yUN,
>
> Thank you for your useful comments and suggestions. We have made significant improvements and additions to our paper based on the feedback received, and would like to draw your attention to [the message we posted jointly to all the reviewers](https://openreview.net/forum?id=Vo1MVffQED&noteId=r7kMrh5H4c). This describes all the improvements made as well as our response to common concerns. We also reply to your specific comments:
> * “The main contribution is to show that, …” We would like to stress that our paper has two main contributions - the theoretical framework as well as the Meta-RL approach. The latter was not emphasized enough and is now made more prominent by giving it its own section. We elaborate this further in the joint message to all reviewers.
> * “the main theorem is not very exciting in some sense.” We refer to the joint message where we address both the novelty of Theorem 1 and the comparison to previous approaches. We particularly want to emphasize that a key ingredient is the way we formulate the problem. This makes the theorem clean to state and is the result of many prior attempts at formalizing this that were more complex but less insightful. The power of Theorem 1 is demonstrated in that it is only using this formulation that we were able to develop the Meta-RL approach, which significantly advances the state of the art in Stackelberg learning. Furthermore, we have added an additional theorem (in the appendix) that shows that without the query-oracle POMDP construction, RL algorithms can provably fail.
> * “the experiments are not particularly impressive” - We have added extensive additional experimental comparisons to existing approaches, as well as an entirely new set of experiments on a novel, much more challenging domain based on Atari 2600 games, which significantly advances the state of the art in Stackelberg learning. Moreover, the games we already used are *iterated* matrix games, not simple one-shot matrix games. This makes them significantly richer and more complex. While they are low-dimensional, they encode highly difficult incentive structures that can be seen as abstractions of many real-world problems. Iterated prisoner’s dilemma in particular is one of the most widely studied social dilemmas in the literature and not an easy problem to solve.
>
> We thank you again for your feedback. Do our paper improvements and additions fully address your concerns? If you have any further questions or comments we would welcome further feedback.

---

> > ### Comment · Reviewer_3yUN · 2022-11-29
> > **Thank you for updated version**
> >
> > Thank you for your updates here. Your clarifications about the theorem are appreciated, and the new experiments, especially on a more complicated setting, do make the paper more convincing. As such, I have increased my score.

---

> ### Comment · Area_Chair_kYyS · 2022-11-21
> **Any comments to the responses from authors?**
>
> Dear Reviewer 3yUN,
>
> Thank you very much for your informative review.  The authors have provided responses to your concerns together with additional experiments.  How did they change your evaluation, particularly on the significance of the main theorem and experimental support?

---

### Official Review · Reviewer_RZA6 · 2022-10-25

**Confidence:** 3
**Correctness:** 3
**Technical Novelty And Significance:** 2
**Empirical Novelty And Significance:** 2
**Recommendation:** 5

**Clarity, Quality, Novelty And Reproducibility:**

The conclusions from theorem 1 seem too strong. Just because some method doesn’t satisfy the conditions of the theorem doesn’t mean it isn’t principled. I could see a lot of approaches working where the leader updates while the follower learns
Theorem 1 also seems pretty obvious and already known from the literature. I think the work should be pretty reproducible. I thought the writing was clear and concise.

**Strength And Weaknesses:**

This paper provides a nice overview of the existing literature for solving sequential Stackelberg games. I especially like table 2. I think that a version of this paper that does a better job experimentally comparing against all the existing approaches and with better theory characterizing when one would expect approaches to not work could be useful to the community.
However, as it currently is, the theory seems to give a result that is already known, the experiments do not sufficiently compare across all the baselines, and the novel algorithmic contribution does not have good enough experiments to back it up.

The experiments should be a lot more extensive and compare against all the related works shown in the table


**Summary Of The Paper:**

This paper combines recent multi-agent RL approaches for finding stackelberg equilibria into one framework, with conditions on convergence. Also, ideas from meta-RL are used to improve the follower.


**Summary Of The Review:**

Overall, I think this paper needs a bit more work, with better theory that proves existing methods won't work, with extensive experiments.

---

> ### Author Response · Authors · 2022-11-18
> **Response to Reviewer RZA6**
>
> Dear Reviewer RZA6,
>
> We would like to thank you for your comments and assessment of our paper, which we have taken as directions to improve our paper. We would like to draw your attention to [the general message we have posted to all reviewers](https://openreview.net/forum?id=Vo1MVffQED&noteId=r7kMrh5H4c), which lists extensive improvements and additions in this revision. This includes new theoretical and experimental results which we hope address your concerns. We will also respond to your individual points below:
> * “a better job experimentally comparing against all the existing approaches”, “the experiments do not sufficiently compare across all the baselines”, “The experiments should be a lot more extensive and compare against all the related works shown in the table” - We have added extensive additional experiments, including a direct comparison to Balaguer et al (to the extent possible given the information they publish) and to Brero et al 2022. Our Meta-RL outperforms both approaches (converging in 50K steps in the iterated prisoners’ dilemma setting, whereas Balaguer et al report final performance after 1.2B steps). It also uses substantially fewer samples than Brero et al, while also achieving higher performance in many of the experiments. These are the only two directly comparable approaches we are aware of; all other approaches listed in Table 2 focus on significantly more restricted settings such as reward shaping.
> * “the novel algorithmic contribution does not have good enough experiments to back it up.” - In addition to the above new comparisons, we have added a new set of experiments on a novel domain that is significantly beyond those tackled by prior Stackelberg approaches. This is a bilateral trade scenario in a modified Atari 2600 game and demonstrates a significant advance in the state of the art in Stackelberg learning. We describe this in more detail in our message to all reviewers. We hope this addresses your concern.
> * “the theory seems to give a result that is already known” - we disagree with this assessment, as our theorem significantly generalizes previous results. This may be partly not-obvious because a lot of the additional power comes from the re-statement of the problem through (arbitrary) follower oracles. But it is precisely this formulation that allows us to develop the Meta-RL approach, for instance, which is the first Stackelberg learning approach we are aware of that moves beyond a typical (and slow) outer-inner loop paradigm. We discuss further novelties compared to prior results in our general message to all reviewers.
> * “with better theory characterizing when one would expect approaches to not work” - We have added a new theoretical result that shows that leader RL can fail without the query-oracle construction in Theorem 1. This is, to our knowledge, the first theoretical proof that RL can fail against best-responding opponents.
> * “The conclusions from Theorem 1 seem too strong. Just because some method doesn’t satisfy the conditions of the theorem doesn’t mean it isn’t principled. I could see a lot of approaches working where the leader updates while the follower learns” Thank you. We agree that principled approaches outside of Theorem 1 are possible, and it was not our intent to claim otherwise. We have now clarified in the discussion after Theorem 1 that the target in Balaguer et al, which is non-Stackelberg, is intentional, and is also a valid optimization target. We have also added a new theorem regarding the necessity of a POMDP construction in the special case of Theorem 1 for leader RL approaches.
> * Regarding “approaches [...] where the leader updates while the follower learns”, this is something we have been considering as well, and while it is an interesting direction, we believe it would be very difficult to give provable guarantees for such an approach, especially without restrictions on the underlying Markov game or the follower learning dynamics. For instance, simply giving reward to the leader during follower learning in a typical outer-inner loop approach can lead to learning failure (we have added an experiment showing this to Appendix C.3.) Similarly, having followers learn continuously - as opposed to re-sampling a random policy after every leader update - can lead to learning getting stuck in a local optimum. (Appendix C.4) While it may well be possible to put additional mechanisms in place to prevent these failures, we believe that proving that such an approach is guaranteed to work would be highly non-trivial (but see this as an interesting avenue for future work).
>
> We again thank you for your valuable feedback which has been immensely helpful in improving our paper. Have we addressed your concerns fully? If you have any more concerns or questions, we would welcome further feedback and discussion.

---

> > ### Comment · Reviewer_RZA6 · 2022-12-06
> > **Reviewer Response**
> >
> > Thanks for making these changes, I think the paper is better now and I have updated my score. However I still recommend reject because the results tie Balaguer et al.

---

> > > ### Author Response · Authors · 2022-12-07
> > > **Response to Reviewer RZA6**
> > >
> > > Dear Reviewer RZA6,
> > >
> > > Thank you for your response and for taking into consideration our paper improvements.
> > >
> > > In regard to the relationship to Balaguer: we assume you’re referring to final performance, and not convergence speed. Yes, final performance is comparable because both approaches achieve at or very near the theoretical optimum in all scenarios. So, this is expected and unsurprising.
> > >
> > > What is pertinent is that our approach gives a drastic improvement in speed of convergence. While due to limited information in Balaguer et al we cannot give an exact number, we note that Balaguer et al report final performance at 1.28 *billion* timesteps, whereas our approach converges in 50 *thousand* timesteps. Balaguer et al is also similar to Brero et al in that it uses an outer-inner loop approach. We show a direct comparison against the latter, and expect that Balaguer et al would look similar.
> > >
> > > Furthermore, note that the new set of experiments we have added on Atari 2600 is significantly more complex than anything Balaguer et al attempt.
> > >
> > > We hope this clears up your concern.

---

> ### Comment · Area_Chair_kYyS · 2022-11-21
> **Any comments to the responses from authors?**
>
> Dear Reviewer RZA6,
>
> Thank you very much for your informative review.  The authors have provided responses to your concerns.  How did they change your evaluation, particularly on the significance of theoretical results and sufficiency of experimental support?

---

### Official Review · Reviewer_FBtF · 2022-10-31

**Confidence:** 3
**Correctness:** 3
**Technical Novelty And Significance:** 3
**Empirical Novelty And Significance:** 2
**Recommendation:** 6

**Clarity, Quality, Novelty And Reproducibility:**

**Clarity**: The paper is very well-written. Every piece of background is clearly and concisely described.

**Novelty**: I believe that the concepts that are mentioned are not very novel. The main idea behind the paper is to propose a unified framework and show that it converges to Stackelberg equilibria. However, I think that the ideas are

**Reproducibility**: The source code to reproduce the experiments is provided in the supplementary materials.

**Strength And Weaknesses:**

Strengths:
- The paper is clearly written, and the concepts are nicely described.
- The background information is described concisely.

------------------------------------------------------------
Weaknesses

- Page 2, last paragraph: "We focus here mainly on this memory-less case." Why do you only consider memory-less cases?
- Page 3, first line: "total reward" ---> this should be "total return"
- The paper is heavily based on three works of Brero et al, which however only one of those is published in an archival venue.

- I would recommend including the payoff matrices of the games in the appendix.

- Page 6, 3rd paragraph: Could you please elaborate more on this sentence: "The leader should receive reward only when followers are playing their best-response equilibrium, and the entire follower learning process should be one episode for the leader." Why do we need such a constraint?

- There are small typos and incorrect use of articles in the paper, which, however, do not make the paper hard to read. For example, on page 8 in the last paragraph: alllow ---> allow

- I would like to see the baselines of Balaguer et al. (2022) included in Figures 1 and 2. Otherwise, the reader has to go to the paper of Balaguer et al. (2022) to see the achieved returns of the baselines.

- I feel that ICLR is not a suitable venue for this paper. ICLR, as its name indicates, focuses on learning representations, and I do not see how this paper fits in this scope. I think conferences such as AAMAS/AAAI/IJCAI/ICML/NeurIPS/COLT are more suitable for this paper.

- Why are the matrix games POMDPs?


**Summary Of The Paper:**

The authors propose a framework for convergence to Stackelberg equilibria in multi-agent RL. The main contribution of the paper can be summarised by Theorem 1 in the paper. Theorem 1 states 3 conditions for convergence so that the solutions for the leader and the follower form a Stackelberg equilibrium. Additionally, 2 more conditions are stated that are required to consider the problem of the leader as a POMDP. Finally, the authors experimentally evaluate when Theorem 1 holds in several matrix games and scenarios.

**Summary Of The Review:**

I recommend a weak reject for this paper. I do not feel that it significantly alters our understanding of Stackelberg equilibria. Additionally, I believe that ICLR is not a suitable venue for this paper. I would expect higher dimensional experiments.
My ranking reflects my current understanding of the paper. I would be happy to change my score if the authors satisfactorily refute my main points of criticism.

---

> ### Author Response · Authors · 2022-11-18
> **Response to Reviewer FBtF (1/2)**
>
> Dear Reviewer FBtF,
>
> Thank you for your detailed and useful comments. We respond to your individual comments below and would also like to draw your attention to [the message we have posted to all the reviewers](https://openreview.net/forum?id=Vo1MVffQED&noteId=r7kMrh5H4c). We want to emphasize that, in addition to Theorem 1, the Meta-RL approach is a second contribution that significantly advances the state of the art on Stackelberg RL. We have also added extensive new theoretical and experimental results, which we also discuss in the joint message.
>
> To your specific points:
> * “Why do you only consider memory-less cases?” To be clear, this only refers to the discussion in the introduction, and is purely to keep notation more concise - the case with memory makes the notation significantly more cumbersome without adding any more insight. However, Theorem 1 does generalize to a leader policy with memory. We discussed this in a paragraph after Theorem 1, now moved to the appendix for space reasons. We have clarified the comment you refer to, to make this clear.
> * Discounted reward vs return: We have changed this to “return” following your suggestion.
> * Brero et al: Our theoretical result significantly generalizes the theorem in Brero et al, in particular to arbitrary follower oracles, which directly enables our Meta-RL approach. Unlike Brero et al we also show that leader RL algorithms can fail in a non-POMDP version. In addition to experiments on this, we have added a new theorem that proves this for some Markov games. The Meta-RL approach is also unrelated to the work of Brero et al.
> * Payoff matrices in the appendix: Yes, we completely agree - thank you! We have added these.
> * Page 6, 3rd paragraph: Essentially, the constraint is that the leader is evaluated against the followers when the followers are best responding. If this condition is violated, this can give the leader an incorrect optimization target. Consider for instance an outer-inner loop approach where followers use Q-learning to compute their best response after every leader policy update (essentially the Brero 2022 approach), and suppose the leader received reward from the Q-learning phase of the followers. Then imagine a setting where the leader has one strategy choice corresponding to a quickly-learnable follower best-response strategy that gives medium reward to the leader, and another leader strategy choice corresponding to a slow-to-learn follower strategy with high leader reward. The latter would be the Stackelberg equilibrium, but the former could give higher reward averaged throughout the learning phase. We have added an additional experiment to Appendix C.3 that demonstrates this. Similarly, if followers’ learning were to terminate early, before they reach their best response, the leader reward can be wrong. It may be possible to construct an approach where followers learn partially, then the leader is evaluated, and then followers continue to learn. However, continuous follower learning in itself can also lead to learning failure (see new experiment in Appendix C.4). We therefore believe that proving such an approach works would be difficult, especially without any restrictions on the underlying Markov game.
> * Typos: Thank you for noticing these, we have fixed them and spell-checked the entire document again.
>
> (message 1 out of 2)

---

> > ### Author Response · Authors · 2022-11-18
> > **Response to Reviewer FBtF (2/2)**
> >
> > (continued from above)
> > * Balaguer et al comparison: We have added this. As Balaguer et al. do not publish learning curves (their figures show a single inner loop iteration at the end of training, not progress over the course of training), we can only show the final performance, reported in Balaguer et al. at 1.28B steps. Our approach converges in 50K environment steps. We have also included a comparison to Brero et al in our iterated matrix game experiments, as well as additional experiments on a novel Atari domain.
> > * ICLR suitability: While we understand your reasoning, we can only point out that ICLR has become much broader in recent years, and as reviewer 671a notes, several Stackelberg learning papers have been published in ICLR.
> > * “Why are the matrix games POMDPs?” Note that these are _iterated_ matrix games. That means the two players play the matrix game several times, and the MDP state is a one-step memory of how each agent behaved in the previous step. For instance, if agent 1 plays action 1, and agent 2 plays action 2, the next state is $s=(1,2)$. This allows for strategies such as “tit for tat” in prisoner’s dilemma, where an agent cooperates if the opponent previously cooperated but retaliates when the opponent defects.
> > * “I do not feel that it significantly alters our understanding of Stackelberg equilibria.” We would like to challenge this by drawing your attention to our Meta-RL approach. This approach is the first Stackelberg approach to go beyond an outer-inner loop paradigm (as used by all previous approaches) and significantly advances our understanding of learning Stackelberg equilibria. It is also only because of our specific statement of Theorem 1 in terms of arbitrary follower oracles (again a novelty) that we were able to develop the Meta-RL approach. Additionally, in a newly added theorem we show that leader RL approaches without the query-oracle POMDP construction can fail. This may have implications for Multi-Agent RL beyond Stackelberg learning. We discuss this also in the note to all the reviewers.
> > * “I would expect higher dimensional experiments.” While iterated matrix games are already challenging state-of-the-art domains for Stackelberg learning, we have added a new set of experiments on a novel environment. In this, we modify an Atari 2600 game to build in a bilateral trade scenario. To our knowledge, his is significantly more complex than any prior Stackelberg domain in the literature and showcases the potential of the Meta-RL approach to drive Stackelberg learning to entirely new kinds of domains. We show that the Meta-RL approach is able to successfully learn two distinct equilibria in this system, depending on which agent is the Stackelberg leader.
> >
> > We would like to thank you again for your detailed and helpful comments, which we hope we have addressed. If there are any remaining questions or concerns please do let us know, we would be grateful for any further feedback that could help us strengthen our paper further.

---

> ### Comment · Area_Chair_kYyS · 2022-11-21
> **Any comments to the responses from authors?**
>
> Dear Reviewer FBtF,
>
> Thank you very much for your detailed review.  The authors have provided responses to your concerns.  How did they change your evaluation particularly on the significance of the contributions?

---

> ### Comment · Reviewer_FBtF · 2022-12-02
> **Acknowledgement of Authors' Response**
>
> I would like to thank the authors for addressing some of my concerns. However, the suitability of ICLR as a venue remains a major concern.
> I have raised my score to 6.

---

> > ### Author Response · Authors · 2022-12-04
> > **Suitability of ICLR**
> >
> > Dear Reviewer FBtF,
> >
> > Thank you for your response and for taking into consideration our paper improvements. Regarding the suitability of ICLR, we would like to offer one additional thought: The Meta-RL approach we introduce can be seen as operating on a representation of the leader behavior. In our current experiments we construct this representation explicitly. But the larger point is that using any (explicit or implicitly learned) representation and generalizing based on this can be a useful approach in learning Stackelberg equilibria, and possibly more broadly in multi-agent RL.
> > A novelty is that whereas previous work on “modeling other agents” have agent A learning to model agent B so that A can learn better, in our case, B (follower) uses a representation of A (leader) so that A can learn better. As far as we know, this is the first work showing that such a representation can be useful for Stackelberg learning, and this new view on the role of representation may also have applications beyond Stackelberg equilibria and could be highly relevant to the ICLR community.

---

### Official Review · Reviewer_671a · 2022-11-05

**Confidence:** 3
**Correctness:** 2
**Technical Novelty And Significance:** 2
**Empirical Novelty And Significance:** 2
**Recommendation:** 3

**Clarity, Quality, Novelty And Reproducibility:**

The writing is hard to follow, and neither the theoretical nor the numerical results are convincing. The reproducibility is okay given the provided codes.

**Strength And Weaknesses:**

Strengths:
* Propose a general framework for implementing Stackelberg equilibria search as a multi-agent RL problem.
* Numerical experiments are provided to demonstrate the usefulness of the proposed framework.

Weaknesses:
* The notation is confusing and inconsistent. For example, in Definition 2, it is unclear how these notation are related to the Markov Game notation in the previous page. Also, it seems that the authors are confusing strategies $s$ with actions $a$. In particular, if ${\bf s}_F$ is the follower strategies, then what is ${\bf a}_F$? Similarly, if $s_L$ is the leader strategy, then what are $s_t$ and $a_L$?
* Theorem 1 is either trivial or very hard to follow, making the claim about a general framework for implementing Stackelberg equilibria search as a multi-agent RL problem unconvincing. The general case seems to be a simple restatement of the problem formulation, and the proof in the appendix also seems to suggest this. The special case is hard to understand. and also $s_L(o)$ and $\mathcal{O}_L$ are undefined. The authors should try to make these statements and proofs more formal, instead of using informal statements like "acting the same during queries as they would in $\mathcal{M}$".
* The authors seem to only compare with Balaguer et al. (2022) as the benchmark. However, there are tons of Stackelberg RL algorithms that the authors fail to compare with. See [A, B] for example.

[A] Tianmin Shu and Yuandong Tian. M3RL: Mind-aware multi-agent management reinforcement learning. In ICLR, 2019.

[B] Runsheng Yu, Xinrun Wang, Youzhi Zhang, Rundong Wang, Bo An, ZhenYu Shi, and Hanjiang Lai. Learning expensive coordination: An event-based deep RL approach. In ICLR, 2020.

**Summary Of The Paper:**

This paper studies Stackelberg equilibria and presents a general framework for implementing Stackelberg equilibria search as a multi-agent RL problem, which encapsulates both previous approaches and some new design spaces such as contextual policies. Some numerical experiments are also provided.

**Summary Of The Review:**

Given the comments above, I think the paper is not qualified for publication in ICLR due to the drawbacks in both writing quality and solidity of the results.

---

> ### Author Response · Authors · 2022-11-18
> **Response to Reviewer 671a**
>
> Dear Reviewer 671a,
>
> We thank you for your very detailed and helpful comments, which have been a great help in improving our paper. We respond to your individual comments below and also draw your attention to [the message we have posted to all reviewers](https://openreview.net/forum?id=Vo1MVffQED&noteId=r7kMrh5H4c). In particular, we want to point out that our paper contains *two* major contributions, the second of which - the Meta-RL algorithm - seems to have been largely overlooked. We discuss this in the general message. We have also made extensive additions to experimental comparisons as well as added an entirely novel evaluation domain (built on Atari 2600), which we believe greatly strengthens our paper and advances the state of the art. Regarding your specific concerns:
> * Regarding notation: In regard to Definition 2: we used $s_L$, $s_F$ to denote the leader and follower policies to emphasize that these are also their strategies in the Markov game, but obviously this clashes with $s_t$ denoting the state at time $t$. This was a mistake on our part, and we have changed the notation in Definition 2 and throughout the paper to use $\pi$ for policies. This is now also in line with Definition 1. Thank you for bringing this to our attention. $s_t$ denotes the state at timestep $t$. $a_F, a_L$ denote the leader and follower action at that timestep. We have changed $a_F$ to $a_{F,t}$ and similarly $a_L$ to $a_{L,t}$ to make this clearer.
> * We also want to emphasize that strategies in Markov games are policies, not actions at a particular step. We are trying to be clear about this distinction throughout the paper. Is there anywhere in particular where you thought we confused strategies and actions? Or is this clear now after we made the notational clarifications?
> * Regarding Theorem 1 notation: As above, $s_L(o)$ was meant to denote the leader’s action given observation $o$, and we have changed this to $\pi_L(o)$. $\mathcal O_L$ was meant to denote the leader’s observation space. We have changed this to $\Omega_L$ in an effort to make the notation more consistent. Again, thank you for bringing these to our attention.
> * Regarding Theorem 1 in general: Please see our message to all reviewers.
> * Regarding the special case of Theorem 1, this resolves a tension between the “whole policy”-level definition of a Stackelberg equilibrium and the “action at each step”-level of reinforcement learning. Intuitively, by including any queries of the leader policy that the follower oracle computation might make into the leader’s rollout, we can transform the equilibrium problem into one at the “action at each step” level.
> Regarding the leader invariance condition: Following your suggestion we have removed this from Theorem 1, as it is automatic in the memory-less case. We now instead discuss this in the section on extending Theorem 1 to leader policies with memory (now in the appendix).
> * Regarding benchmark comparisons: We have added a more detailed comparison to Balaguer et al, and also a comparison to Brero et al.
> We also thank you for your literature suggestions, which we have added. Note however that despite the similar name, these papers solve a very different problem than we do. The type of “Stackelberg Markov games” that both of these papers address is a significant restriction compared to the general Markov games our paper deals with. They also both use at least partially rule-based followers. We also note that neither paper formally proves a Stackelberg equilibrium guarantee.
>
> Again, we want to thank you for your immensely helpful comments and suggestions. Please let us know if we have addressed your concerns, and if there is anything we can do to strengthen our paper further.

---

> ### Comment · Area_Chair_kYyS · 2022-11-21
> **Any comments to the responses from authors?**
>
> Dear Reviewer 671a,
>
> Thank you very much for your informative review.  The authors have provided responses to your concerns.  How did they change your evaluation, particularly on clarity, theorem 1, and baselines?

---

> > ### Comment · Reviewer_671a · 2022-11-27
> > **Comments to the responses from authors**
> >
> > Dear AC and authors,
> >
> > I would like to thank the authors for the detailed reply and revision of the paper draft. The notation issues have been fixed as far as I'm concerned. However, the two other major issues have not been addressed as far as I understand.
> > * For the baselines, the restriction of [A, B] to special cases of Stackelberg games is not a convincing excuse for excluding them from the numerical comparisons, especially given that the current paper is proposing a general framework to handle a generic follower oracle. In particular, it's important to allow for rule-based and bounded rational followers, instead of just followers leading to exact equilibria.
> > * For Theorem 1, I still find the statement and the result for the special case ambiguous and hard to follow. I think there are two major issues that the authors need to further address.
> >    * The first is that the concepts of oracle and query oracle are ambiguous, and the authors should clearly define the mathematical definition of an oracle, and particularly the input and output of it. This also makes the proof in the appendix hard to understand due to the related concepts of "internal states", "queries" and "initial segments" that are not rigorously explained/defined in the main text, to name just a few.
> >       * Also, based on the description of the paper, the oracle seems to be yet another kind of "rule", which further validates the necessity to compare with baseline Stackelberg RL algorithms requiring rule-based followers, like [A, B].
> >       * Some example of oracles and queries and query oracles should also be given to help the readers understand the concepts. In addition, the authors should clearly provide some examples that are not "query oracle cases".
> >    * The second is that the statement that the leader's problem is a POMDP in the special case is too abstract in the main text. Since the authors propose algorithms based on this theorem, a constructive and explicit POMDP formulation should be stated in the main text clearly instead as part of the proof in the appendix. The authors should also give explicit examples on how a Stackelberg game with a query oracle for followers is written into a POMDP. Otherwise it's hard to tell what this theorem is really about from a rigorous viewpoint (and even whether it's correct or not).
> >
> > So based on these, I decide to maintain my current score.

---

> > > ### Author Response · Authors · 2022-11-29
> > > **Response to Reviewer 671a**
> > >
> > > Dear Reviewer 671a,
> > >
> > > Thank you for your additional useful feedback. We would like to respond below.
> > >
> > > * Regarding prior works [A,B], we want to clarify that these solve a *different* problem than we do. The Stackelberg games in  [A,B] are a reward-shaping problem, whereas we focus on settings where both leader and follower act in the environment together (such as for instance in our new Atari-based experiments). The authors of [A] themselves state, verbatim, that these games are “Different from the common Markov game setting for MARL in prior work” (page 3 in [A]). Given that [A,B] solve a different problem, we do not see how a direct comparison would be possible. For [B] we are also unable to find source code.
> > >   * In regard to generality: while our Theorem 1 statement is general, we do not claim that the meta-RL algorithm, which is one possible instantiation of an algorithm that is consistent with the theorem, will be suitable to all settings (e.g., some settings may require more than contextual policies).
> > > * Regarding Theorem 1: We thank you for your detailed comments, we now better understand the issues and will clarify these in the paper.
> > >   * Regarding the oracle and query oracle concepts: By “oracle” we mean any algorithm that implements the followers’ best response to a given leader policy. We define this just before Definition 2, where we first mention $\mathcal E(\mathcal F_{\pi_L})$. It takes as input the leader policy $\pi_L$, and computes as output follower policies $\pi_F$ for all followers, or a distribution over follower policies. Based on your comments, we will make this a formal definition, and also change the notation slightly to $\mathcal E_{\mathcal M} (\pi_L)$ to further emphasize the input.
> > >   * The special case of Theorem 1 requires that the oracle implementation only needs query access to the leader policy $\pi_L$, i.e., only uses the leader policy to compute values $\pi_L(o_i)$ for some sequence of inputs $(o_i)$. This is in contrast to implementations that have access to a description of $\pi_L$, e.g., to the weights of a neural network if that is how the leader policy is implemented. You are right that “query oracle” is not formally defined in the paper, because we only use the term informally to refer to the special case of Theorem 1. We will clarify this in the paper.
> > >   * We will add concrete examples to the paper, thank you for this suggestion.
> > >     * Most practical approaches would only need query access - for instance, a typical learning algorithm for the followers would fall in that category, and similarly our Meta-RL approach characterizes the leader policy through queries.
> > >     * A description-based oracle implementation could apply, for example, to linear leader policies: If $\theta_{s,a}$ describes the probability of the leader taking action $a$ in state $s$, this could be used to directly compute a best-response without sampling actions from the leader policy. By the general case of Theorem 1, this would work, for instance, with  evolutionary strategies for the leader. But because this oracle implementation does not use queries, we can not use the POMDP construction of the special case of Theorem 1 to make this work with RL-based leaders. Indeed in the new theorem in the Appendix we show that an approach that operates without queries can fail together with an RL-based leader  even on a small domain.
> > >   * You are right to point out that rule-based followers can be seen as a type of best-response oracle implementation, and we thank you for this observation. We will add this to the paper.
> > >   * We will move the construction of the leader POMDP to the main text, and make the construction more explicit.
> > >   * Does the above clarify the Theorem statement?

---

### Author Response · Authors · 2022-11-18
**Response to All Reviewers**

Dear Reviewers,

In addition to individual responses, we respond below to some common concerns, and discuss the major improvements and additions we have made to the paper based on your helpful feedback. In the following, we clarify the contributions of our paper, explain the significance of Theorem 1, detail a new theoretical result we’ve added, discuss additional experimental comparisons to prior work, and report a new set of experiments on a novel domain that is well beyond the prior state of the art.
* First, we want to clarify that there are two main contributions in our paper: Theorem 1, and the Meta-RL algorithm. The latter seems to largely have been overlooked. We elaborate on this below, and have made this much clearer in the paper.
## Theorem 1
* A key ingredient in the theorem statement is the formulation through follower oracles. This framing makes the theorem clean to state and is the result of many prior attempts at formalizing the same intuition.
* Regarding Theorem 1 vs Brero et al, out result is more general in several ways:
  * We generalize to arbitrary follower oracles, which directly enables our Meta-RL approach.
  * We generalize to the non-POMDP case, which is relevant to non-RL leader algorithms.
  * We demonstrate in a new theorem that RL approaches without the query-oracle construction can fail (see below).
  * We give a proof for arbitrary Markov games.
* To our knowledge, Theorem 1 is the first theoretical result for Stackelberg learning in arbitrary Markov games. All previous approaches were for specific settings, and / or sequential only for the leader or followers, not both (i.e., not full Markov games).
* We added a theorem proving that without the query-oracle construction of Theorem 1, RL algorithms will diverge even on a small domain. To our knowledge, a divergence of RL against best-responding opponents has not been shown before, and this could have relevance beyond Stackelberg learning.
## Meta-RL approach
Regarding the Meta-RL algorithm, we view this as an important contribution and this is given new prominence in the revision by giving it its own section.
* This approach allows us to break out of the outer-loop inner-loop paradigm. To our knowledge this is the first Stackelberg learning approach that achieves this. There is no inner loop as the meta-follower can best-respond without further training.
* This provides considerable speedup, beating existing approaches by orders of magnitude in sample complexity in experiments. For instance, we converge in 50K timesteps, whereas Balaguer et al. report results after 1.28B time steps (We had previously misunderstood the number of environment steps in the Balaguer et al paper. This is now fixed.)
* The speedup of our approach is so substantial that it enables entirely new kinds of domains (see new experiments below).
* This is also the first work using Meta-RL as a means to deal with changing behavior in other agents, in itself a major novelty. This likely has implications beyond Stackelberg equilibria.
## New Experimental Domain
We added a new set of experiments on an entirely novel and highly challenging domain. This is based on a bilateral trade scenario built on a modified Atari 2600 game and is orders of magnitude more complex than anything previously described in the Stackelberg learning literature. Our Meta-RL approach is able to exactly learn two distinct Stackelberg equilibria in this setting. We believe these experiments demonstrate a significant advance in the state of art in Stackelberg learning.
## Existing Experiments
It is difficult to make direct comparisons to prior approaches, in part because most do not deal with as general a case as this present paper, as detailed in the literature review and Table 2. The two approaches that are comparable are Brero et al and Balaguer et al. Of these, the latter does not currently make source code available, and in our view does not give enough detail to allow a faithful reimplementation. However, we understand that this is not satisfactory. We have thus made extensive changes and additions to our paper:
* We make a direct comparison with Balaguer et al by including their published performance in Figure 1. This is their final performance (they do not publish learning curves) at 1.28 billion environment timesteps. Our approach converges in around 50 thousand timesteps.
* We have reimplemented the Brero et al 2022 approach. The Meta-RL approach drastically outperforms it both in terms of achieved performance and sample efficiency, the latter by orders of magnitude.
* On the use of iterated matrix games: we choose these because they are among the most challenging, existing Stackelberg benchmark domains. While these domains are low-dimensional, some of them have extremely challenging incentive structures. Iterated prisoner’s dilemma in particular is one of the most widely studied, and considered one of the most difficult, social dilemma problems.

---

> ### Author Response · Authors · 2022-11-18
> **Full list of changes to the paper**
>
> For reference, we include a full list of changes we made to the revised paper:
> * New additions:
>   * We have added performance comparisons to the PPO+Q-learn approach from Brero et al and to Balaguer et al.
>   * We have added an additional and entirely novel evaluation domain using a bilateral trade scenario on a modified Atari 2600 game.
>   * We have added a formal proof that without the query-oracle POMDP construction, RL algorithms will necessarily diverge on some Markov games.
>   * We have added several new experiments to test the conditions of Theorem 1.
> * Restructuring:
>   * We have moved the experiments testing the specific conditions of Theorem 1 into the appendix.
>   * We have moved the discussion of our Meta-RL approach into its own section
>   * We moved the discussion of leader memory into the appendix.
> * Smaller changes:
>   * We removed the leader invariance condition from the main theorem, as in the memory-less case this is automatic, and as per reviewer comments this is confusing. We instead discuss it in the context of stateful leader policies in the appendix.
>   * As a counterpoint to Theorem 1, we added a short theorem of necessary conditions for Stackelberg learning in the appendix, informing possible avenues for future work.
>   * We added details on the new Atari experiments to the appendix.
>   * We added Table 3, with all the payoff matrices used.
>   * We changed s_L, s_F to pi_L, pi_F throughout the paper to denote leader and follower strategy or policy, and made other small notational fixes.
>   * We clarified that the focus on the memory-less case is only for concise notation during the introduction and that the main theoretical result generalizes to a setting with leader memory.
>   * We changed “total reward” to “total return”
>   * We made various other small fixes (typos, formatting, reviewer suggestions)

---

### Decision · Program_Chairs · 2023-01-20

**Decision:**

Reject

**Justification For Why Not Higher Score:**

The novelty and implications of the main theorem is unclear.

**Justification For Why Not Lower Score:**

N/A

**Metareview: Summary, Strengths And Weaknesses:**

This paper studies the problem of computing a Stackelberg equilibrium and proves the conditions for this problem to be formulated as a partially observable Markov decision process (POMDP).  The main theorem essentially states that the leader policy at a Stackelberg equilibrium is the optimal policy for a POMDP given the access to the best response of followers.  Motivated by this theorem, the paper proposes a reinforcement learning method for computing the Stackelberg equilibrium, where the key difference from the prior work is that the proposed approach ignores the samples when the followers are not best responding.  The effectiveness of the proposed approach is empirically validated with a small normal form game and a strategic game of a modified Atari environment.

The strength of the paper is in the general framework that unifies existing approaches to computing Stackelberg equilibria.  However, it is unclear what new insights that Theorem 1 gives beyond the state-of-the-art knowledge about the Stackelberg equilibrium.  Since the general case directly follows from the definition, what is essential is the Query-Oracle Special Case. While it has been improved during discussion, further clarifications and details are needed to precisely understand the implications of the theorem.

* Following is a personal opinion of the area chair:

Reviewer 671a suggested that "a constructive and explicit POMDP formulation should be stated in the main text", but I find that the POMDP formulation in the proof in the appendix is unclear.  This is partly because it is unclear what can be observed by each party.  I suggest writing the POMDP formulation in more detail and with more clarity.